# ENFORCING ROBUST CONTROL GUARANTEES WITHIN NEURAL NETWORK POLICIES

**Priya L. Donti[1], Melrose Roderick[1], Mahyar Fazlyab[2], J. Zico Kolter[1,3]**
[1]Carnegie Mellon University, [2]Johns Hopkins University, [3]Bosch Center for AI
{pdonti, mroderick}@cmu.edu, mahyarfazlyab@jhu.edu, zkolter@cs.cmu.edu

## ABSTRACT

When designing controllers for safety-critical systems, practitioners often face a challenging tradeoff between robustness and performance. While robust control methods provide rigorous guarantees on system stability under certain worst-case disturbances, they often yield simple controllers that perform poorly in the average (non-worst) case. In contrast, nonlinear control methods trained using deep learning have achieved state-of-the-art performance on many control tasks, but often lack robustness guarantees. In this paper, we propose a technique that combines the strengths of these two approaches: constructing a generic nonlinear control policy class, parameterized by neural networks, that nonetheless enforces the same provable robustness criteria as robust control. Specifically, our approach entails integrating custom convex-optimization-based projection layers into a neural network-based policy. We demonstrate the power of this approach on several domains, improving in average-case performance over existing robust control methods and in worst-case stability over (non-robust) deep RL methods.

## 1 INTRODUCTION

The field of robust control, dating back many decades, has been able to provide rigorous guarantees on when controllers will succeed or fail in controlling a system of interest. In particular, if the uncertainties in the underlying dynamics can be bounded in specific ways, these techniques can produce controllers that are provably robust even under worst-case conditions. However, as the resulting policies tend to be simple (i.e., often linear), this can limit their performance in typical (rather than worst-case) scenarios. In contrast, recent high-profile advances in deep reinforcement learning have yielded state-of-the-art performance on many control tasks, due to their ability to capture complex, nonlinear policies. However, due to a lack of robustness guarantees, these techniques have still found limited application in safety-critical domains where an incorrect action (either during training or at runtime) can substantially impact the controlled system.

In this paper, we propose a method that combines the guarantees of robust control with the flexibility of deep reinforcement learning (RL). Specifically, we consider the setting of nonlinear, time-varying systems with unknown dynamics, but where (as common in robust control) the uncertainty on these dynamics can be bounded in ways amenable to obtaining provable performance guarantees. Building upon specifications provided by traditional robust control methods in these settings, we construct a new class of nonlinear policies that are parameterized by neural networks, but that are nonetheless *provably robust*. In particular, we *project* the outputs of a nominal (deep neural network-based) controller onto a space of stabilizing actions characterized by the robust control specifications. The resulting nonlinear control policies are trainable using standard approaches in deep RL, yet are *guaranteed* to be stable under the same worst-case conditions as the original robust controller.

We describe our proposed deep nonlinear control policy class and derive efficient, differentiable projections for this class under various models of system uncertainty common in robust control. We demonstrate our approach on several different domains, including synthetic linear differential inclusion (LDI) settings, the cart-pole task, a quadrotor domain, and a microgrid domain. Although these domains are simple by modern RL standards, we show that purely RL-based methods often produce unstable policies in the presence of system disturbances, both during and after training. In contrast, we show that our method remains stable even when worst-case disturbances are present, while improving upon the performance of traditional robust control methods.

## 2 RELATED WORK

We employ techniques from robust control, (deep) RL, and differentiable optimization to learn provably robust nonlinear controllers. We discuss these areas of work in connection to our approach.

**Robust control.** Robust control is concerned with the design of feedback controllers for dynamical systems with modeling uncertainties and/or external disturbances (Zhou and Doyle, 1998; Başar and Bernhard, 2008), specifically controllers with guaranteed performance under worst-case conditions. Many classes of robust control problems in both the time and frequency domains can be formulated using linear matrix inequalities (LMIs) (Boyd et al., 1994; Kothare et al., 1996); for reasonably-sized problems, these LMIs can be solved using off-the-shelf numerical solvers based on interior-point or first-order (gradient-based) methods. However, providing stability guarantees often requires the use of simple (linear) controllers, which greatly limits average-case performance. Our work seeks to improve performance via *nonlinear* controllers that nonetheless retain the same stability guarantees.

**Reinforcement learning (RL).** In contrast, RL (and specifically, deep RL) is not restricted to simple controllers or problems with uncertainty bounds on the dynamics. Instead, deep RL seeks to learn an optimal control policy, represented by a neural network, by directly interacting with an unknown environment. These methods have shown impressive results in a variety of complex control tasks (e.g., Mnih et al. (2015); Akkaya et al. (2019)); see Buşoniu et al. (2018) for a survey. However, due to its lack of safety guarantees, deep RL has been predominantly applied to simulated environments or highly-controlled real-world problems, where system failures are either not costly or not possible.

Efforts to address the lack of safety and stability in RL fall into several main categories. The first tries to combine control-theoretic ideas, predominantly robust control, with the nonlinear control policy benefits of RL (e.g., Morimoto and Doya (2005); Abu-Khalaf et al. (2006); Feng et al. (2009); Liu et al. (2013); Wu and Luo (2013); Luo et al. (2014); Friedrich and Buss (2017); Pinto et al. (2017); Jin and Lavaei (2018); Chang et al. (2019); Han et al. (2019); Zhang et al. (2020)). For example, RL has been used to address stochastic stability in $H_\infty$ control synthesis settings by jointly learning Lyapunov functions and policies in these settings (Han et al., 2019). As another example, RL has been used to address $H_\infty$ control for continuous-time systems via min-max differential games, in which the controller and disturbance are the "minimizer" and "maximizer" (Morimoto and Doya, 2005). We view our approach as thematically aligned with this previous work, though our method is able to capture not only $H_\infty$ settings, but also a much broader class of robust control settings.

Another category of methods addressing this challenge is safe RL, which aims to learn control policies while maintaining some notion of safety during or after learning. Typically, these methods attempt to restrict the RL algorithm to a safe region of the state space by making strong assumptions about the smoothness of the underlying dynamics, e.g., that the dynamics can be modeled as a Gaussian process (GP) (Turchetta et al., 2016; Akametalu et al., 2014) or are Lipschitz continuous (Berkenkamp et al., 2017; Wachi et al., 2018). This framework is in theory more general than our approach, which requires using stringent uncertainty bounds (e.g. state-control norm bounds) from robust control. However, there are two key benefits to our approach. First, norm bounds or polytopic uncertainty can accommodate sharp discontinuities in the continuous-time dynamics. Second, convex projections (as used in our method) scale polynomially with the state-action size, whereas GPs in particular scale exponentially (and are therefore difficult to extend to high-dimensional problems).

A third category of methods uses Constrained Markov Decision Processes (C-MDPs). These methods seek to maximize a discounted reward while bounding some discounted cost function (Altman, 1999; Achiam et al., 2017; Taleghan and Dietterich, 2018; Yang et al., 2020). While these methods do not require knowledge of the cost functions a-priori, they only guarantee the cost constraints hold during test time. Additionally, using C-MDPs can yield other complications, such as optimal policies being stochastic and the constraints only holding for a subset of states.

**Differentiable optimization layers.** A great deal of recent work has studied differentiable optimization layers for neural networks: e.g., layers for quadratic programming (Amos and Kolter, 2017), SAT solving (Wang et al., 2019), submodular optimization (Djolonga and Krause, 2017; Tschiatschek et al., 2018), cone programs (Agrawal et al., 2019), and other classes of optimization problems (Gould et al., 2019). These layers can be used to construct neural networks with useful inductive bias for particular domains or to enforce that networks obey hard constraints dictated by the settings in which they are used. We create fast, custom differentiable optimization layers for the latter purpose, namely, to project neural network outputs into a set of certifiably stabilizing actions.

## 3   BACKGROUND ON LQR AND ROBUST CONTROL SPECIFICATIONS

In this paper, our aim is to control nonlinear (continuous-time) dynamical systems of the form

$$\dot{x}(t) \in A(t)x(t) + B(t)u(t) + G(t)w(t), \tag{1}$$

where $x(t) \in \mathbb{R}^s$ denotes the state at time $t$; $u(t) \in \mathbb{R}^a$ is the control input; $w(t) \in \mathbb{R}^d$ captures both external (possibly stochastic) disturbances and any modeling discrepancies; $\dot{x}(t)$ denotes the time derivative of the state $x$ at time $t$; and $A(t) \in \mathbb{R}^{s \times s}, B(t) \in \mathbb{R}^{s \times a}, G(t) \in \mathbb{R}^{s \times d}$. This class of models is referred to as linear differential inclusions (LDIs); however, we note that despite the name, this class does indeed characterize *nonlinear* systems, as, e.g., $w(t)$ can depend arbitrarily on $x(t)$ and $u(t)$ (though we omit this dependence in the notation for brevity). Within this class of models, it is often possible to construct robust control specifications certifying system stability. Given such specifications, our proposal is to learn nonlinear (deep neural network-based) policies that *provably* satisfy these specifications while optimizing some objective of interest. We start by giving background on the robust control specifications and objectives considered in this work.

### 3.1   ROBUST CONTROL SPECIFICATIONS

In the continuous-time, infinite-horizon settings we consider here, the goal of robust control is often to construct a time-invariant control policy $u(t) = \pi(x(t))$, alongside some certification that guarantees that the controlled system will be stable (i.e., that trajectories of the system will converge to an equilibrium state, usually $x = 0$ by convention; see Haddad and Chellaboina (2011) for a more formal definition). For many classes of systems,[1] this certification is typically in the form of a positive definite Lyapunov function $V : \mathbb{R}^s \to \mathbb{R}$, with $V(0) = 0$ and $V(x) > 0$ for all $x \neq 0$, such that the function is decreasing along trajectories – for instance,

$$\dot{V}(x(t)) \leq -\alpha V(x(t)) \tag{2}$$

for some design parameter $\alpha > 0$. (This particular condition implies *exponential stability* with a rate of convergence $\alpha$.[2]) For certain classes of bounded dynamical systems, time-invariant linear control policies $u(t) = Kx(t)$, and quadratic Lyapunov functions $V(x) = x^T P x$, it is possible to construct such guarantees using semidefinite programming. For instance, consider the class of norm-bounded LDIs (NLDIs)

$$\dot{x} = Ax(t) + Bu(t) + Gw(t), \quad \|w(t)\|_2 \leq \|Cx(t) + Du(t)\|_2, \tag{3}$$

where $A \in \mathbb{R}^{s \times s}, B \in \mathbb{R}^{s \times a}, G \in \mathbb{R}^{s \times d}, C \in \mathbb{R}^{k \times s}$, and $D \in \mathbb{R}^{k \times a}$ are time-invariant and known, and the disturbance $w(t)$ is arbitrary (and unknown) but obeys the norm bounds above.[3] For these systems, it is possible to specify a set of stabilizing policies via a set of linear matrix inequalities (LMIs, Boyd et al. (1994)):

$$\begin{bmatrix} AS + SA^T + \mu GG^T + BY + Y^T B^T + \alpha S & SC^T + Y^T D^T \\ CS + DY & -\mu I \end{bmatrix} \preceq 0, \;\; S \succ 0, \;\; \mu > 0, \tag{4}$$

where $S \in \mathbb{R}^{s \times s}$ and $Y \in \mathbb{R}^{a \times s}$. For matrices $S$ and $Y$ satisfying (4), $K = YS^{-1}$ and $P = S^{-1}$ are then a stabilizing linear controller gain and Lyapunov matrix, respectively. While the LMI above is specific to NLDI systems, this general paradigm of constructing stability specifications using LMIs applies to many settings commonly considered in robust control (e.g., settings with norm-bounded disturbances or polytopic uncertainty, or $H_\infty$ control settings). More details about these types of formulations are given in, e.g., Boyd et al. (1994); in addition, we provide the relevant LMI constraints for the settings we consider in this work in Appendix A.

---

[1]In this work, we consider sub-classes of system (1) that may indeed be stochastic (e.g., due to a stochastic external disturbance $w(t)$), but that can be bounded so as to be amenable to deterministic stability analysis. However, other settings may require stochastic stability analysis; please see Astrom (1971).

[2]See, e.g., Haddad and Chellaboina (2011) for a more rigorous definition of (local and global) exponential stability. Condition (2) comes from *Lyapunov's Theorem*, which characterizes various notions of stability using Lyapunov functions.

[3]A slightly more complex formulation involves an additional term in the norm bound, i.e., $Cx(t) + Du(t) + Hw(t)$, which creates a quadratic inequality in $w$. The mechanics of obtaining robustness specifications in this setting are largely the same as presented here, though with some additional terms in the equations. As such, as is often done, we assume that $H = 0$ for simplicity.

## 3.2 LQR CONTROL OBJECTIVES

In addition to designing for stability, it is often desirable to optimize some objective characterizing controller performance. While our method can optimize performance with respect to any arbitrary cost or reward function, to make comparisons with existing methods easier, for this paper we consider the well-known infinite-horizon "linear-quadratic regulator" (LQR) cost, defined as

$$\int_0^\infty \left( x(t)^T Q x(t) + u(t)^T R u(t) \right) dt, \tag{5}$$

for some $Q \in \mathbb{S}^{s \times s} \succeq 0$ and $R \in \mathbb{S}^{a \times a} \succ 0$. If the control policy is assumed to be time-invariant and linear as described above (i.e., $u(t) = Kx(t)$), minimizing the LQR cost subject to stability constraints can be cast as an SDP (see, e.g., Yao et al. (2001)) and solved using off-the-shelf numerical solvers – a fact that we exploit in our work. For example, to obtain an optimal linear time-invariant controller for the NLDI systems described above, we can solve

$$\underset{S,Y}{\text{minimize}} \quad \text{tr}(QS) + \text{tr}(R^{1/2} Y S^{-1} Y^T R^{1/2}) \quad \text{s.t. Equation (4) holds.} \tag{6}$$

## 4 ENFORCING ROBUST CONTROL GUARANTEES WITHIN NEURAL NETWORKS

We now present the main contribution of our paper: A class of *nonlinear* control policies, potentially parameterized by deep neural networks, that is guaranteed to obey the same stability conditions enforced by the robustness specifications described above. The key insight of our approach is as follows: While it is difficult to derive specifications that globally characterize the stability of a generic nonlinear controller, if we are given *known* robustness specifications, we can create a sufficient condition for stability by simply enforcing that our policy satisfies these specifications at all $t$. For instance, given a known Lyapunov function, we can enforce exponential stability by ensuring that our policy sufficiently decreases this function (e.g., satisfies Equation (2)) at any given $x(t)$.

In the following sections, we present our nonlinear policy class, as well as our general framework for learning provably robust policies using this policy class. We then derive the instantiation of this framework for various settings of interest. In particular, this involves constructing (custom) differentiable projections that can be used to adjust the output of a nominal neural network to satisfy desired robustness criteria. For simplicity of notation, we will often suppress the $t$-dependence of $x$, $u$, and $w$, but we note that these are continuous-time quantities as before.

### 4.1 A PROVABLY ROBUST NONLINEAR POLICY CLASS

Given a dynamical system of the form (1) and a quadratic Lyapunov function $V(x) = x^T P x$, let

$$\mathcal{C}(x) := \{ u \in \mathbb{R}^a \mid \dot{V}(x) \leq -\alpha V(x) \quad \forall \dot{x} \in A(t)x + B(t)u + G(t)w \} \tag{7}$$

denote a set of actions that, for a *fixed* state $x \in \mathbb{R}^s$, are guaranteed to satisfy the exponential stability condition (2) (even under worst-case realizations of the disturbance $w$). We note that this "safe" set is non-empty if $P$ satisfies the relevant LMI constraints (e.g., system (4) for NLDIs) characterizing robust linear time-invariant controllers, as there is then some $K$ corresponding to $P$ such that $Kx \in \mathcal{C}(x)$ for all states $x$.

Using this set of actions, we then construct a robust nonlinear policy class that *projects* the output of some neural network onto this set. More formally, consider an arbitrary nonlinear (neural network-based) policy class $\hat{\pi}_\theta : \mathbb{R}^s \rightarrow \mathbb{R}^a$ parameterized by $\theta$, and let $\mathcal{P}_{(\cdot)}$ denote the projection operator for some set $(\cdot)$. We then define our robust policy class as $\pi_\theta : \mathbb{R}^s \rightarrow \mathbb{R}^a$, where

$$\pi_\theta(x) = \mathcal{P}_{\mathcal{C}(x)}(\hat{\pi}_\theta(x)). \tag{8}$$

We note that this policy class is differentiable if the projections can be implemented in a differentiable manner (e.g., using convex optimization layers (Agrawal et al., 2019), though we construct efficient custom solvers for our purposes). Importantly, as all policies in this class satisfy the stability condition (2) for all states $x$ and at all times $t$, these policies are *certifiably* robust under the same conditions as the original (linear) controller for which the Lyapunov function $V(x)$ was constructed.

Given this policy class and some performance objective $\ell$ (e.g., LQR cost), our goal is to then find parameters $\theta$ such that the corresponding policy optimizes this objective – i.e., to solve

$$\underset{\theta}{\text{minimize}} \quad \int_0^\infty \ell \left( x, \pi_\theta(x) \right) dt \quad \text{s.t.} \quad \dot{x} \in A(t)x + B(t)\pi_\theta(x) + G(t)w. \tag{9}$$

---

**Algorithm 1** Learning provably robust controllers with deep RL

---

1: **input** performance objective $\ell$     *// e.g., LQR cost*
2: **input** stability requirement     *// e.g., $\dot{V}(x) \leq -\alpha V(x)$*
3: **input** policy optimizer $\mathcal{A}$     *// e.g., a planning or RL algorithm*
4: **compute** $P$, $K$ satisfying LMI constraints   *// e.g., by optimizing (6)*
5: **construct** specifications $\mathcal{C}(x)$ using $P$   *// as defined in Equation (7)*
6: **construct** robust policy class $\pi_\theta$ using $\mathcal{C}$   *// as defined in Equation (8)*
7: **train** $\pi_\theta$ via $\mathcal{A}$ to optimize Equation (9)
8: **return** $\pi_\theta$

---

Since $\pi_\theta$ is differentiable, we can solve this problem via a variety of approaches, e.g., a model-based planning algorithm if the true dynamics are known, or virtually any (deep) RL algorithm if the dynamics are unknown.[4]

This general procedure for constructing stabilizing controllers is summarized in Algorithm 1. While seemingly simple, this formulation presents a powerful paradigm: by simply transforming the output of a neural network, we can employ an expressive policy class to optimize an objective of interest while *ensuring* the resultant policy will stabilize the system during both training and testing.

We instantiate our framework by constructing "safe" sets $\mathcal{C}(x)$ and their associated (differentiable) projections $\mathcal{P}_{\mathcal{C}(x)}$ for three settings of interest: NLDIs, polytopic linear differential inclusions (PLDIs), and $H_\infty$ control settings. As an example, we describe this procedure below for NLDIs, and refer readers to Appendix B for corresponding formulations for the additional settings we consider.

### 4.2 EXAMPLE: NLDIS

In order to apply our framework to the NLDI setting (3), we first compute a quadratic Lyapunov function $V(x) = x^T P x$ by solving the optimization problem (6) for the given system via semidefinite programming. We then use the resultant Lyapunov function to compute the system-specific "safe" set $\mathcal{C}(x)$, and then create a fast, custom differentiable solver to project onto this set.

#### 4.2.1 COMPUTING SETS OF STABILIZING ACTIONS

Given $P$, we compute $\mathcal{C}_{\text{NLDI}}(x)$ as the set of actions $u \in \mathbb{R}^a$ that, for each state $x \in \mathbb{R}^s$, satisfy the stability condition (2) at that state *under even a worst-case realization of the dynamics* (i.e., in this case, even under a worst-case disturbance $w$). The form of the resultant set is given below.

**Theorem 1.** *Consider the NLDI system* (3)*, some stability parameter $\alpha > 0$, and a Lyapunov function $V(x) = x^T P x$ with $P$ satisfying Equation* (4)*. Assuming $P$ exists, define*

$$\mathcal{C}_{\text{NLDI}}(x) := \left\{ u \in \mathbb{R}^a \mid \|Cx + Du\|_2 \leq \frac{-x^T PB}{\|G^T Px\|_2} u - \frac{x^T (2PA + \alpha P)x}{2\|G^T Px\|_2} \right\}$$

*for all states $x \in \mathbb{R}^s$. For all $x$, $\mathcal{C}_{\text{NLDI}}(x)$ is a non-empty set of actions that satisfy the exponential stability condition* (2)*. Further, $\mathcal{C}_{\text{NLDI}}(x)$ is a convex set in $u$.*

*Proof.* We seek to find a set of actions such that the condition (2) is satisfied along *all* possible trajectories of (3). A set of actions satisfying this condition at a given $x$ is given by

$$\mathcal{C}_{\text{NLDI}}(x) := \left\{ u \in \mathbb{R}^a \mid \sup_{w : \|w\|_2 \leq \|Cx + Du\|_2} \dot{V}(x) \leq -\alpha V(x) \right\}.$$

Let $\mathcal{S} := \{w : \|w\|_2 \leq \|Cx + Du\|_2\}$. We can then rewrite the left side of the above inequality as

$$\sup_{w \in \mathcal{S}} \dot{V}(x) = \sup_{w \in \mathcal{S}} \dot{x}^T P x + x^T P \dot{x} = 2x^T P(Ax + Bu) + \sup_{w \in \mathcal{S}} 2x^T PGw$$

$$= 2x^T P(Ax + Bu) + 2\|G^T Px\|_2 \|Cx + Du\|_2,$$

by the definition of the NLDI dynamics and the closed-form minimization of a linear term over an $L_2$ ball. Rearranging yields an inequality of the desired form. We note that by definition of the

---

[4]While this problem is infinite-horizon and continuous in time, in practice, one would optimize it in discrete time over a large finite time horizon.

---

specifications (4), there is some $K$ corresponding to $P$ such that the policy $u = Kx$ satisfies the exponential stability condition (2); thus, $Kx \in \mathcal{C}_{\text{NLDI}}$, and $\mathcal{C}_{\text{NLDI}}$ is non-empty. Further, as the above inequality represents a second-order cone constraint in $u$, this set is convex in $u$. □

We further consider the special case where $D = 0$, i.e., the norm bound on $w$ does not depend on the control action. This form of NLDI arises in many common settings (e.g., where $w$ characterizes linearization error in a nonlinear system but the dynamics depend only linearly on the action), and is one for which we can compute the relevant projection in closed form (as described shortly).

**Corollary 1.1.** *Consider the NLDI system* (3) *with $D = 0$, some stability parameter $\alpha > 0$, and Lyapunov function $V(x) = x^T P x$ with $P$ satisfying Equation* (4)*. Assuming $P$ exists, define*

$$\mathcal{C}_{\text{NLDI-0}}(x) := \left\{ u \in \mathbb{R}^a \mid 2x^T PBu \leq -x^T(2PA + \alpha P)x - 2\|G^T Px\|_2\|Cx\|_2 \right\}$$

*for all states $x \in \mathbb{R}^s$. For all $x$, $\mathcal{C}_{\text{NLDI-0}}(x)$ is a non-empty set of actions that satisfy the exponential stability condition* (2)*. Further, $\mathcal{C}_{\text{NLDI-0}}(x)$ is a convex set in $u$.*

*Proof.* The result follows by setting $D = 0$ in Theorem 1 and rearranging terms. As the above inequality represents a linear constraint in $u$, this set is convex in $u$. □

### 4.2.2 DERIVING EFFICIENT, DIFFERENTIABLE PROJECTIONS

For the general NLDI setting (3), we note that the relevant projection $\mathcal{P}_{\mathcal{C}_{\text{NLDI}}(x)}$ (see Theorem 1) represents a projection onto a second-order cone constraint. As this projection does not necessarily have a closed form, we must implement it using a differentiable optimization solver (e.g., Agrawal et al. (2019)). For computational efficiency purposes, we implement a custom solver that employs an accelerated projected dual gradient method for the forward pass, and employs implicit differentiation through the fixed point equations of this solution method to compute relevant gradients for the backward pass. Derivations and additional details are provided in Appendix C.

In the case where $D = 0$ (see Corollary 1.1), we note that the projection operation $\mathcal{P}_{\mathcal{C}_{\text{NLDI-0}}(x)}$ does have a closed form, and can in fact be implemented via a single ReLU operation. Specifically, defining $\eta^T := 2x^T PB$ and $\zeta := -x^T(2PA + \alpha P)x - 2\|G^T Px\|_2\|Cx\|_2$, we see that

$$\mathcal{P}_{\mathcal{C}_{\text{NLDI-0}}(x)}(\hat{\pi}(x)) = \begin{cases} \hat{\pi}(x) & \text{if } \eta^T \hat{\pi}(x) \leq \zeta \\ \hat{\pi}(x) - \frac{\eta^T \hat{\pi}(x) - \zeta}{\eta^T \eta}\eta & \text{otherwise} \end{cases} = \hat{\pi}(x) - \text{ReLU}\left(\frac{\eta^T \hat{\pi}(x) - \zeta}{\eta^T \eta}\right)\eta. \quad (10)$$

## 5 EXPERIMENTS

Having instantiated our general framework, we demonstrate the power of our approach on a variety of simulated control domains.[5] In particular, we evaluate performance on the following metrics:

- **Average-case performance:** How well does the method optimize the performance objective (i.e., LQR cost) under average (non-worst case) dynamics?

- **Worst-case stability:** Does the method remain stable even when subjected to adversarial (worst-case) dynamics?

In all cases, we show that our method is able to improve performance over traditional robust controllers under average conditions, while still guaranteeing stability under worst-case conditions.

### 5.1 DESCRIPTION OF DYNAMICS SETTINGS

We evaluate our approach on five NLDI settings: two synthetic NLDI domains, the cart-pole task, a quadrotor domain, and a microgrid domain. (Additional experiments for PLDI and $H_\infty$ control settings are described in Appendix I.) For each setting, we choose a time discretization based on the speed at which the system evolves, and run each episode for 200 steps over this discretization. In all cases except the microgrid setting, we use a randomly generated LQR objective where the matrices $Q^{1/2}$ and $R^{1/2}$ are drawn i.i.d. from a standard normal distribution.

---

[5]Code for all experiments is available at `https://github.com/locuslab/robust-nn-control`

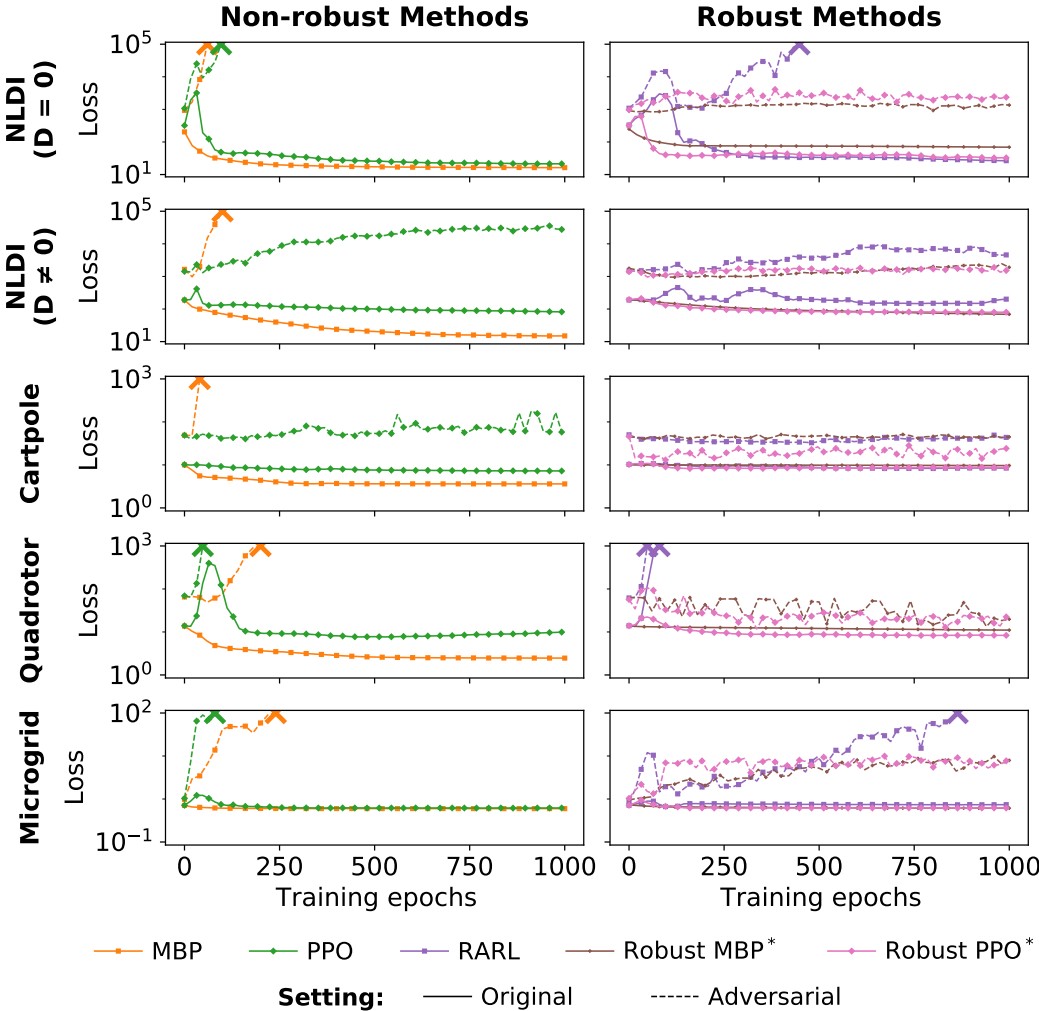

Figure 1: Test performance over training epochs for all learning methods employed in our experiments. For each training epoch (10 updates for the MBP model and 18 for PPO), we report average quadratic loss over 50 episodes, and use "X" to indicate cases where the relevant method became unstable. (Lower loss is better.) Our robust methods (denoted by *), unlike the non-robust methods and RARL, remain stable under adversarial dynamics throughout training.

**Synthetic NLDI settings.** We generate NLDIs of the form (3) with $s = 5$, $a = 3$, and $d = k = 2$ by generating matrices $A, B, G, C$ and $D$ i.i.d. from normal distributions, and producing the disturbance $w(t)$ using a randomly-initialized neural network (with its output scaled to satisfy the norm-bound on the disturbance). We investigate settings both where $D \neq 0$ and where $D = 0$. In both cases, episodes are run for 2 seconds at a discretization of 0.01 seconds.

**Cart-pole.** In the cart-pole task, our goal is to balance an inverted pendulum resting on top of a cart by exerting horizontal forces on the cart. For our experiments, we linearize this system as an NLDI with $D \neq 0$ (see Appendix D), and add a small additional randomized disturbance satisfying the NLDI bounds. Episodes are run for 10 seconds at a discretization of 0.05 seconds.

**Planar quadrotor.** In this setting, our goal is to stabilize a quadcopter in the two-dimensional plane by controlling the amount of force provided by the quadcopter's right and left thrusters. We linearize this system as an NLDI with $D = 0$ (see Appendix E), and add a small disturbance as in the cart-pole setting. Episodes are run for 4 seconds at a discretization of 0.02 seconds.

**Microgrid.** In this final setting, we aim to stabilize a microgrid by controlling a storage device and a solar inverter. We augment the system given in Lam et al. (2016) with LQR matrices and NLDI bounds (see Appendix F). Episodes are run for 2 seconds at a discretization of 0.01 seconds.

## 5.2 EXPERIMENTAL SETUP

We demonstrate our approach by constructing a robust policy class (8) for each of these settings, and optimizing this policy class via different approaches. Specifically, we construct a nominal nonlinear control policy class as $\hat{\pi}_\theta(x) = Kx + \tilde{\pi}_\theta(x)$, where $K$ is obtained via robust LQR optimization (6), and where $\tilde{\pi}_\theta(x)$ is a feedforward neural network. To construct the projections $\mathcal{P}_C$, we employ the value of $P$ obtained when solving for $K$. For the purposes of demonstration, we then optimize our robust policy class $\pi_\theta(x) = \mathcal{P}_C(\hat{\pi}_\theta(x))$ using two different methods:

- **Robust MBP** (ours)**:** A model-based planner that assumes the true dynamics are known.

- **Robust PPO** (ours)**:** An RL approach based on PPO (Schulman et al., 2017) that does not assume known dynamics (beyond the bounds used to construct the robust policy class).

Robust MBP is optimized using gradient descent for 1,000 updates, where each update samples 20 roll-outs. Robust PPO is trained for 50,000 updates, where each update samples 8 roll-outs; we choose the model that performs best on a hold-out set of initial conditions during training. We note that while we use PPO for our demonstration, our approach is agnostic to the particular method of training, and can be deployed with many different (deep) RL paradigms.

We compare our robust neural network-based method against the following baselines:

- **Robust LQR:** Robust (linear) LQR controller obtained via Equation (6).

- **Robust MPC:** A robust model-predictive control algorithm (Kothare et al., 1996) based on state-dependent LMIs. (As the relevant LMIs are not always guaranteed to solve, our implementation temporarily reverts to the Robust LQR policy when that occurs.)

- **RARL:** The robust adversarial reinforcement learning algorithm (Pinto et al., 2017), which trains an RL agent in the presence of an adversary. (We note that unlike the other robust methods considered here, this method is not *provably* robust.)

- **LQR:** A standard non-robust (linear) LQR controller.

- **MBP** and **PPO:** The non-robust neural network policy class $\hat{\pi}_\theta(x)$ optimized via a model-based planner and the PPO algorithm, respectively.

In order to evaluate performance, we *train* all methods on the dynamical settings described in Section 5.1, and *evaluate* them on two different variations of the dynamics:

- **Original dynamics:** The dynamical settings described above ("average case").

- **Adversarial dynamics:** Modified dynamics with an adversarial test-time disturbance $w(t)$ generated to maximize loss ("worst case"). We generate this disturbance separately for each method described above (see Appendix G for more details).

Initialization states are randomly generated for all experiments. For the synthetic NLDI and microgrid settings, these are generated from a standard normal distribution. For both cart-pole and quadrotor, because our NLDI bounds model linearization error, we must generate initial points within a region where this linearization holds. In particular, the linearization bounds only hold for a specified $L_\infty$ ball, $B_{\text{NLDI}}$, around the equilibrium. We use a simple heuristic to construct this ball and jointly find a smaller $L_\infty$ ball, $B_{\text{init}}$, such that there exists a level set $L$ of the Robust LQR Lyapunov function with $B_{\text{init}} \subseteq L \subseteq B_{\text{NLDI}}$ (details in Appendix H). Since Robust LQR (and by extension our methods) are guaranteed to decrease the relevant Lyapunov function, this guarantees that these methods will never leave $B_{\text{NLDI}}$ when initialized starting from any point inside $B_{\text{init}}$ – i.e., that our NLDI bounds will always hold throughout the trajectories produced by these methods.

## 5.3 RESULTS

Table 1 shows the performance of the above methods. We report the integral of the quadratic loss over the prescribed time horizon on a test set of states, or indicate cases where the relevant method became unstable (i.e., the loss became orders of magnitude larger than for other approaches). (Sample trajectories for these methods are also provided in Appendix H.)

These results illustrate the basic advantage of our approach. In particular, both our Robust MBP and Robust PPO methods show **improved "average-case" performance over the other provably robust methods** (namely, Robust LQR and Robust MPC). As expected, however, the non-robust

| Environment | | LQR | MBP | PPO | Robust LQR | Robust MPC | RARL | Robust MBP* | Robust PPO* |
|---|---|---|---|---|---|---|---|---|---|
| Generic NLDI | O | 373 | **16** | 21 | 253 | 253 | **27** | 69 | 33 |
| ($D = 0$) | A | ———— *unstable* ———— | | | 1009 | 873 | *unstable* | 1111 | 2321 |
| Generic NLDI | O | 278 | **15** | 82 | 199 | 199 | 147 | **69** | 80 |
| ($D \neq 0$) | A | ———— *unstable* ———— | | | 1900 | 1667 | *unstable* | 1855 | 1669 |
| Cart-pole | O | 36.3 | **3.6** | 7.2 | 10.2 | 10.2 | **8.3** | 9.7 | 8.4 |
| | A | — *unstable* — | | 172.1 | 42.2 | 47.8 | 41.2 | 50.0 | 16.3 |
| Quadrotor | O | 5.4 | **2.5** | 7.7 | 13.8 | 13.8 | 12.2 | 11.0 | **8.3** |
| | A | *unstable* | 545.7 | 137.6 | 64.8 | *unstable*† | 63.1 | 25.7 | 26.5 |
| Microgrid | O | 4.59 | **0.60** | 0.61 | 0.73 | 0.73 | 0.67 | **0.61** | **0.61** |
| | A | ———— *unstable* ———— | | | 0.99 | 0.92 | 2.17 | 7.68 | 8.91 |

Table 1: Performance of various approaches, both robust (right) and non-robust (left). We report average quadratic loss over 50 episodes under the original dynamics (O) and under an adversarial disturbance (A). For the original dynamics (O), the best performance for both non-robust methods and robust methods is in bold (lower loss is better). Under the adversarial dynamics (A), we seek to observe whether or not methods remain stable; we use "*unstable*" to indicate cases where the relevant method becomes unstable (and † to denote any instabilities due to numerical, rather than theoretical, issues). Our robust methods (denoted by *) improve performance over Robust LQR and Robust MPC in the average case while remaining stable under adversarial dynamics, whereas the non-robust methods and RARL either go unstable or receive much larger losses.

LQR, MBP, and PPO methods often perform better within the original nominal dynamics, as they are optimizing for expected performance but do not need to consider robustness under worst-case scenarios. However, when we apply allowable adversarial perturbations (that still respect our disturbance bounds), the non-robust LQR, MBP, and PPO approaches diverge or perform very poorly. Similarly, the RARL agent performs well under the original dynamics, but diverges under adversarial perturbations in the generic NLDI settings. In contrast, both of our provably robust approaches (as well as Robust LQR) **remain stable under even "worst-case" adversarial dynamics**. (We note that the baseline Robust MPC method goes unstable in one instance, though this is due to numerical instability issues, rather than issues with theoretical guarantees.)

Figure 1 additionally shows the performance of all neural network-based methods on the test set over training epochs. While the robust and non-robust MBP and PPO approaches both converge quickly to their final performance levels, both non-robust versions become unstable under the adversarial dynamics very early in the process. The RARL method also frequently destabilizes during training. Our Robust MBP and PPO policies, on the other hand, **remain stable throughout the *entire* optimization process**, i.e., do not destabilize during either training *or* testing. Overall, these results show that our method is able to learn policies that are more expressive than traditional robust methods, while *guaranteeing* these policies will be stable under the same conditions as Robust LQR.

## 6 CONCLUSION

In this paper, we have presented a class of nonlinear control policies that combines the expressiveness of neural networks with the provable stability guarantees of traditional robust control. This policy class entails projecting the output of a neural network onto a set of stabilizing actions, parameterized via robustness specifications from the robust control literature, and can be optimized using a model-based planning algorithm if the dynamics are known or virtually any RL algorithm if the dynamics are unknown. We instantiate our general framework for dynamical systems characterized by several classes of linear differential inclusions that capture many common robust control settings. In particular, this entails deriving efficient, differentiable projections for each setting, via implicit differentiation techniques. We show over a variety of simulated domains that our method improves upon traditional robust LQR techniques while, unlike non-robust LQR and neural network methods, remaining stable even under worst-case allowable perturbations of the underlying dynamics.

We believe that our approach highlights the possible connections between traditional control methods and (deep) RL methods. Specifically, by enforcing more structure in the classes of deep networks we consider, it is possible to produce networks that *provably* satisfy many of the constraints that have typically been thought of as outside the realm of RL. We hope that this work paves the way for future approaches that can combine more structured uncertainty or robustness guarantees with RL, in order to improve performance in settings traditionally dominated by classical robust control.

ACKNOWLEDGMENTS

This work was supported by the Department of Energy Computational Science Graduate Fellowship (DE-FG02-97ER25308), the Center for Climate and Energy Decision Making through a cooperative agreement between the National Science Foundation and Carnegie Mellon University (SES-00949710), the Computational Sustainability Network, and the Bosch Center for AI. This material is based upon work supported by the National Science Foundation Graduate Research Fellowship Program under Grant No. DGE1745016. Any opinions, findings, and conclusions or recommendations expressed in this material are those of the author(s) and do not necessarily reflect the views of the National Science Foundation.

We thank Vaishnavh Nagarajan, Filipe de Avila Belbute Peres, Anit Sahu, Asher Trockman, Eric Wong, and anonymous reviewers for their feedback on this work.

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

# A  DETAILS ON ROBUST CONTROL SPECIFICATIONS

As described in Section 3.1, for many dynamical systems of the form (1), it is possible to specify a set of linear, time-invariant policies guaranteeing infinite-horizon exponential stability via a set of LMIs. Here, we derive the LMI (4) provided in the main text for the NLDI system (3), and additionally describe relevant LMI systems for systems characterized by polytopic linear differential inclusions (PLDIs) and for $H_\infty$ control settings.

## A.1  EXPONENTIAL STABILITY IN NLDIS

Consider the general NLDI system (3). We seek to design a time-invariant control policy $u(t) = Kx(t)$ and a quadratic Lyapunov function $V(x) = x^T P x$ with $P \succ 0$ for this system that satisfy the exponential stability criterion $\dot{V}(x) \leq -\alpha V(x)$, $\forall t$. We derive an LMI characterizing such a controller and Lyapunov function, closely following and expanding upon the derivation provided in Boyd et al. (1994).

Specifically, consider the NLDI system (3), reproduced below:

$$\dot{x} = Ax + Bu + Gw, \quad \|w\|_2 \leq \|Cx + Du\|_2. \tag{A.1}$$

The time derivative of this Lyapunov function along the trajectories of the closed-loop system is

$$\begin{aligned}
\dot{V}(x) &= \dot{x}^T P x + x^T P \dot{x} \\
&= (Ax + Bu + Gw)^T P x + x^T P(Ax + Bu + Gw) \\
&= ((A + BK)x + Gw)^T P x + x^T P((A + BK)x + Gw) \\
&= \begin{bmatrix} x \\ w \end{bmatrix}^T \begin{bmatrix} (A + BK)^T P + P(A + BK) & PG \\ G^T P & 0 \end{bmatrix} \begin{bmatrix} x \\ w \end{bmatrix}.
\end{aligned} \tag{A.2}$$

The exponential stability condition $\dot{V}(x) \leq -\alpha V(x)$ is thus implied by inequality

$$\begin{bmatrix} x \\ w \end{bmatrix}^T M_1 \begin{bmatrix} x \\ w \end{bmatrix} := \begin{bmatrix} x \\ w \end{bmatrix}^T \begin{bmatrix} (A + BK)^T P + P(A + BK) + \alpha P & PG \\ G^T P & 0 \end{bmatrix} \begin{bmatrix} x \\ w \end{bmatrix} \leq 0. \tag{A.3}$$

Additionally, the norm bound on $w$ can be equivalently expressed as

$$\begin{bmatrix} x \\ w \end{bmatrix}^T M_2 \begin{bmatrix} x \\ w \end{bmatrix} := \begin{bmatrix} x \\ w \end{bmatrix}^T \begin{bmatrix} (C + DK)^T(C + DK) & 0 \\ 0 & -I \end{bmatrix} \begin{bmatrix} x \\ w \end{bmatrix} \geq 0. \tag{A.4}$$

Using the S-procedure, it follows that for some $\lambda \geq 0$, the following matrix inequality is a sufficient condition for exponential stability:

$$M_1 + \lambda M_2 \preceq 0. \tag{A.5}$$

Using Schur Complements, this matrix inequality is equivalent to

$$(A + BK)^T P + P(A + BK) + \alpha P + \lambda(C + DK)^T(C + DK) + \frac{1}{\lambda} PGG^T P \preceq 0. \tag{A.6}$$

Left- and right-multiplying both sides by $P^{-1}$, and making the change of variables $S = P^{-1}$, $Y = KS$, and $\mu = 1/\lambda$, we obtain

$$SA^T + AS + Y^T B^T + BY + \alpha S + \frac{1}{\mu} \left(SC^T + Y^T D^T\right)(CS + DY) + \mu GG^T \preceq 0. \tag{A.7}$$

Using Schur Complements again on this inequality, we obtain our final system of linear matrix inequalities as

$$\begin{bmatrix} AS + SA^T + \mu GG^T + BY + Y^T B^T + \alpha S & SC^T + Y^T D^T \\ CS + DY & -\mu I \end{bmatrix} \preceq 0, \quad S \succ 0, \quad \mu > 0, \tag{A.8}$$

where then $K = YS^{-1}$ and $P = S^{-1}$. Note that the first matrix inequality is homogeneous; we can therefore assume $\mu = 1$ (and therefore, $\lambda = 1$), without loss of generality.

## A.2 Exponential stability in PLDIs

Consider the setting of polytopic linear differential inclusions (PLDIs), where the dynamics are of the form

$$\dot{x}(t) = A(t)x(t) + B(t)u(t), \quad (A(t), B(t)) \in \text{Conv}\{(A_1, B_1), \ldots, (A_L, B_L)\}. \quad (A.9)$$

Here, $A(t) \in \mathbb{R}^{s \times s}$ and $B(t) \in \mathbb{R}^{s \times a}$ can vary arbitrarily over time, as long as they lie in the convex hull (denoted $\text{Conv}$) of the set of points above, where $A_i \in \mathbb{R}^{s \times s}$, $B_i \in \mathbb{R}^{s \times a}$ for $i = 1, \ldots, L$.

We seek to design a time-invariant control policy $u(t) = Kx(t)$ and quadratic Lyapunov function $V(x) = x^T P x$ with $P \succ 0$ for this system that satisfy the exponential stability criterion $\dot{V}(x) \leq -\alpha V(x)$, $\forall t$. Such a controller and Lyapunov function exist if there exist $S \in \mathbb{R}^{s \times s} \succ 0$ and $Y \in \mathbb{R}^{a \times s}$ such that

$$A_i S + B_i Y + S A_i^T + Y^T B_i^T + \alpha S \preceq 0, \quad \forall i = 1, \ldots, L, \quad (A.10)$$

where then $K = YS^{-1}$ and $P = S^{-1}$. The derivation of this LMI follows similarly to that for exponential stability in NLDIs, and is well-described in Boyd et al. (1994).

## A.3 $H_\infty$ control

Consider the following $H_\infty$ control setting with linear time-invariant dynamics

$$\dot{x}(t) = Ax(t) + Bu(t) + Gw(t), \quad w \in \mathcal{L}_2, \quad (A.11)$$

where $A$, $B$, and $G$ are time-invariant as for the NLDI case, and where we define $\mathcal{L}_2$ as the set of time-dependent signals with finite $\mathcal{L}_2$ norm.[6]

In cases such as these with larger or more unstructured disturbances, it may not be possible to guarantee asymptotic convergence to an equilibrium. In these cases, our goal is to construct a robust controller with bounds on the extent to which disturbances affect some performance output (e.g., LQR cost), as characterized by the $\mathcal{L}_2$ gain of the disturbance-to-output map. Specifically, we consider the stability requirement that this $\mathcal{L}_2$ gain be bounded by some parameter $\gamma > 0$ when disturbances are present, and that the system be exponentially stable in the disturbance-free case. This requirement can be characterized via the condition that for all $t$ and some $\sigma \geq 0$,

$$\mathcal{E}(x, \dot{x}, u) := \dot{V}(x) + \alpha V(x) + \sigma \left( x^T Q x + u^T R u - \gamma^2 \|w\|_2^2 \right) \leq 0. \quad (A.12)$$

We note that when $\mathcal{E}(x(t), \dot{x}(t), u(t)) \leq 0$ for all $t$, both of our stability criteria are met. To see this, note that integrating both sides of (A.12) from 0 to $\infty$ and ignoring the non-negative terms on the left hand side after integration yields

$$\int_0^\infty (x(t)^T Q x(t) + u(t)^T R u(t)) dt \leq \gamma^2 \int_0^\infty \|w(t)\|_2^2 dt + (1/\sigma) V(x(0)). \quad (A.13)$$

This is precisely the desired bound on the $\mathcal{L}_2$ gain of the disturbance-to-output map (see Khalil and Grizzle (2002)). We also note that in the disturbance-free case, substituting $w = 0$ into (A.12) yields

$$\dot{V}(x) \leq -\alpha V(x) - \sigma \left( x^T Q x + u^T R u \right) \leq -\alpha V(x), \quad (A.14)$$

where the last inequality follows from the non-negativity of the LQR cost; this is precisely our condition for exponential stability.

We now seek to design a time-invariant control policy $u(t) = Kx(t)$ and quadratic Lyapunov function $V(x) = x^T P x$ with $P \succ 0$ that satisfies the above condition. In particular, we can write

$$\mathcal{E}\left( x(t), (A + BK)x(t) + Gw(t), Kx(t) \right) = \begin{bmatrix} x(t) \\ w(t) \end{bmatrix}^T M_1 \begin{bmatrix} x(t) \\ w(t) \end{bmatrix}, \quad (A.15)$$

where

$$M_1 := \begin{bmatrix} (A + BK)^T P + P(A + BK) + \alpha P + \sigma(Q + K^T R K) & PG \\ G^T P & -\gamma^2 \sigma I \end{bmatrix}. \quad (A.16)$$

---

[6]The $\mathcal{L}_2$ norm of a time-dependent signal $w(t) \colon [0, \infty) \to \mathbb{R}^d$ is defined as $\sqrt{\int_0^\infty \|w(t)\|_2^2 dt}$.

Therefore, we seek to find a $P \in \mathbb{R}^{s \times s} \succ 0$ and $K \in \mathbb{R}^{s \times a}$ that satisfy $M_1 \preceq 0$, for some design parameters $\alpha > 0$ and $\sigma > 0$. Using Schur complements, the matrix inequality $M_1 \preceq 0$ is equivalent to

$$(A + BK)^T P + P(A + BK) + \alpha P + \sigma(Q + K^T RK) + PGG^T P/(\gamma^2 \sigma) \preceq 0. \quad \text{(A.17)}$$

As in Appendix A.1, we left- and right-multiply both sides by $P^{-1}$, and make the change of variables $S = P^{-1}$, $Y = KS$, and $\mu = 1/\sigma$ to obtain

$$SA^T + AS + Y^T B^T + BY + \alpha S + \frac{1}{\mu}\left((SQ^{1/2})(Q^{1/2}S) + (Y^T R^{1/2})(R^{1/2}Y)\right) + \mu GG^T/\gamma^2 \preceq 0.$$

Using Schur Complements again, we obtain the LMI

$$\begin{bmatrix} SA^T + AS + Y^T B^T + BY + \alpha S + \mu GG^T/\gamma^2 & \begin{bmatrix} SQ^{1/2} & Y^T R^{1/2} \end{bmatrix} \\ \begin{bmatrix} Q^{1/2}S \\ R^{1/2}Y \end{bmatrix} & -\mu I \end{bmatrix} \preceq 0, \quad S \succ 0, \quad \mu > 0, \tag{A.18}$$

where then $K = YS^{-1}$, $P = S^{-1}$, and $\sigma = 1/\mu$.

# B   DERIVATION OF SETS OF STABILIZING POLICIES AND ASSOCIATED PROJECTIONS

We describe the construction of the set of actions $\mathcal{C}(x)$, defined in Equation (7), for PLDI systems (A.9) and $H_\infty$ control settings (A.11). (The relevant formulations for the NLDI system (3) are described in the main text.)

## B.1   EXPONENTIAL STABILITY IN PLDIS

For the general PLDI system (A.9), relevant sets of exponentially stabilizing actions $\mathcal{C}_{\text{PLDI}}$ are given by the following theorem.

**Theorem B.1.** *Consider the PLDI system* (A.9), *some stability parameter $\alpha > 0$, and a Lyapunov function $V(x) = x^T Px$ with $P$ satisfying* (A.10). *Assuming $P$ exists, define*

$$\mathcal{C}_{PLDI}(x) := \left\{ u \in \mathbb{R}^a \,\middle|\, \begin{bmatrix} 2x^T PB_1 \\ 2x^T PB_2 \\ \vdots \\ 2x^T PB_L \end{bmatrix} u \leq - \begin{bmatrix} x^T(\alpha P + 2PA_1)x \\ x^T(\alpha P + 2PA_2)x \\ \vdots \\ x^T(\alpha P + 2PA_L)x \end{bmatrix} \right\}$$

*for all states $x \in \mathbb{R}^s$. For all $x$, $\mathcal{C}_{PLDI}(x)$ is a non-empty set of actions that satisfy the exponential stability condition* (2). *Further, $\mathcal{C}_{PLDI}(x)$ is a convex set in $u$.*

*Proof.* We seek to find a set of actions such that the condition (2) is satisfied along *all* possible trajectories of (A.9), i.e., for *any* allowable instantiation of $(A(t), B(t))$. A set of actions satisfying this condition at a given $x$ is given by

$$\mathcal{C}_{\text{PLDI}}(x) := \{u \in \mathbb{R}^a \mid \dot{V}(x) \leq -\alpha V(x) \; \forall (A(t), B(t)) \in \text{Conv}\{(A_1, B_1), \ldots, (A_L, B_L)\}.$$

Expanding the left side of the inequality above, we see that for some coefficients $\gamma_i \in \mathbb{R} \geq 0, i = 1, \ldots, L$ satisfying $\sum_{i=1}^L \gamma_i(t) = 1$,

$$\dot{V}(x) = \dot{x}^T Px + x^T P\dot{x} = 2x^T P(A(t)x + B(t)u)$$

$$= 2x^T P\left(\sum_{i=1}^L \gamma_i(t)A_i x + \gamma_i(t)B_i u\right) = \sum_{i=1}^L \gamma_i\left(2x^T P(A_i x + B_i u)\right)$$

by definition of the PLDI dynamics and of the convex hull. Thus, if we can ensure

$$2x^T P(A_i x + B_i u) \leq -\alpha V(x) = -\alpha x^T Px, \; \forall i = 1, \ldots, L,$$

then we can ensure that exponential stability holds. Rearranging this condition and writing it in matrix form yields an inequality of the desired form. We note that by definition of the specifications (A.10), there is some $K$ corresponding to $P$ such that the policy $u = Kx$ satisfies all of the above inequalities; thus, $Kx \in \mathcal{C}_{\text{PLDI}}(x)$, and $\mathcal{C}_{\text{PLDI}}(x)$ is non-empty. Further, as the above inequality represents a linear constraint in $u$, this set is convex in $u$. □

We note that the relevant projection $\mathcal{P}_{\mathcal{C}_{\mathrm{PLDI}}(x)}$ represents a projection onto an intersection of halfspaces, and can thus be implemented via differentiable quadratic programming (Amos and Kolter, 2017).

## B.2 $H_\infty$ CONTROL

For the $H_\infty$ control system (A.11), relevant sets of actions satisfying the condition (A.12) are given by the following theorem.

**Theorem B.2.** *Consider the system* (A.11)*, some stability parameter $\alpha > 0$, and a Lyapunov function $V(x) = x^T P x$ with $P$ satisfying Equation* (A.18)*. Assuming $P$ exists, define*

$$\mathcal{C}_{H_\infty}(x) := \left\{ u \in \mathbb{R}^a \mid u^T R u + (2B^T P x)^T u + x^T \left( P A + A^T P + \alpha P + Q + \gamma^{-2} P G G^T P \right) x \le 0 \right\}$$

*for all states $x \in \mathbb{R}^s$. For all $x$, $\mathcal{C}_{H_\infty}(x)$ is a non-empty set of actions that guarantee condition* (A.12)*, i.e., that the $\mathcal{L}_2$ gain of the disturbance-to-output map is bounded by $\gamma$ and that the system is exponentially stable in the disturbance-free case. Further, $\mathcal{C}_{H_\infty}(x)$ is convex in $u$.*

*Proof.* We seek to find a set of actions such that the condition $\mathcal{E}(x, \dot{x}, u) \le 0$ is satisfied along *all* possible trajectories of (A.11), where $\mathcal{E}$ is defined as in (A.12). A set of actions satisfying this condition at a given $x$ is given by

$$\mathcal{C}_{H_\infty}(x) := \{ u \in \mathbb{R}^a \mid \sup_{w \in \mathcal{L}_2} \mathcal{E}(x, \dot{x}, u) \le 0, \ \dot{x} = Ax + Bu + Gw \}.$$

To begin, we note that

$$\begin{aligned} \mathcal{E}(x, Ax + Bu + Gw, u) = {} & x^T P(Ax + Bu + Gw) + (Ax + Bu + Gw)^T P x + \alpha x^T P x \\ & + \sigma \left( x^T Q x + u^T R u - \gamma^2 \|w\|_2^2 \right) \end{aligned}$$

We then maximize $\mathcal{E}$ over $w$:

$$w^\star = \arg\max_w \mathcal{E}(x, Ax + Bu + Gw, u) = G^T P x / (\sigma \gamma^2). \tag{B.1}$$

Therefore,

$$\mathcal{C}_{H_\infty}(x) = \{ u \mid \mathcal{E}(x, Ax + Bu + Gw^\star, u, w^\star) \le 0 \}. \tag{B.2}$$

Expanding and rearranging terms, this becomes

$$\mathcal{C}_{\mathcal{H}_\infty}(x) = \{ u \mid u^T (\sigma R) u + (2B^T P x)^T u + x^T \left( PA + A^T P + \alpha P + \sigma Q + P G G^T P / (\sigma \gamma^2) \right) x \le 0 \}. \tag{B.3}$$

We note that by definition of the specifications (A.18), there is some $K$ corresponding to $P$ such that the policy $u = Kx$ satisfies the conditions above (see (A.17)); thus, $Kx \in \mathcal{C}_{H_\infty}$, and $\mathcal{C}_{H_\infty}$ is non-empty. We note further that $\mathcal{C}_{\mathcal{H}_\infty}$ is an ellipsoid in the control action space, and is thus convex in $u$. $\qquad\square$

We rewrite the set $\mathcal{C}_{H_\infty}(x)$ such that the projection $\mathcal{P}_{\mathcal{C}_{H_\infty}(x)}$ can be viewed as a second-order cone projection, in order to leverage our fast custom solver (Appendix C). In particular, defining $\tilde{P} = \sigma R$, $\tilde{q} = B^T P x$, and $\tilde{r} = x^T \left( PA + A^T P + \alpha P + \sigma Q + P G G^T P / (\sigma \gamma^2) \right) x$, we can rewrite the ellipsoid above as

$$\mathcal{C}_{H_\infty}(x) = \{ u \mid u^\top \tilde{P} u + 2\tilde{q}^\top u + \tilde{r} \le 0 \}. \tag{B.4}$$

We note that as $\tilde{P} \succ 0$ and $\tilde{r} - \tilde{q}^\top \tilde{P}^{-1} \tilde{q} < 0$, this ellipsoid is non-empty (see, e.g., section B.1 in Boyd and Vandenberghe (2004)). We can then rewrite the ellipsoid as

$$\mathcal{C}_{H_\infty}(x) = \{ u \mid \|\tilde{A}u + \tilde{b}\|_2 \le 1 \} \tag{B.5}$$

where $\tilde{A} = \sqrt{\frac{\tilde{P}}{\tilde{q}^\top \tilde{P}^{-1} \tilde{q} - \tilde{r}}}$ and $\tilde{b} = \sqrt{\frac{P}{q^\top P^{-1} q - r}} P^{-1} q$. The constraint $\|\tilde{A}u + \tilde{b}\|_2 \le 1$ is then a second-order cone constraint in $u$.

## C A FAST, DIFFERENTIABLE SOLVER FOR SECOND-ORDER CONE PROJECTION

In order to construct the robust policy class described in Section 4 for the general NLDI system (3) and the $H_\infty$ setting (A.11), we must project a nominal (neural network-based) policy onto the second-order cone constraints described in Theorem 1 and Appendix B.2, respectively. As this projection operation does not necessarily have a closed form, we implement it via a custom differentiable optimization solver.

More generally, consider a set of the form

$$\mathcal{C} = \{x \in \mathbb{R}^n \mid \|Ax + b\|_2 \le c^T x + d\} \tag{C.1}$$

for some $A \in \mathbb{R}^{m \times n}$, $b \in \mathbb{R}^m$, $c \in \mathbb{R}^n$, and $d \in \mathbb{R}$. Given some input $y \in \mathbb{R}^n$, we seek to compute the second-order cone projection $\mathcal{P}_\mathcal{C}(y)$ by solving the problem

$$\begin{aligned} \underset{x \in \mathbb{R}^n}{\text{minimize}} \quad & \frac{1}{2}\|x - y\|_2^2 \\ \text{subject to} \quad & \|Ax + b\|_2 \le c^T x + d. \end{aligned} \tag{C.2}$$

Let $\mathcal{F}$ denote the $\ell_2$ norm cone, i.e., $\mathcal{F} := \{(w, t) \mid \|w\|_2 \le t\}$. Introducing the auxiliary variable $z \in \mathbb{R}^{m+1}$, we can then rewrite the above optimization problem equivalently as

$$\begin{aligned} \underset{x \in \mathbb{R}^n,\, z \in \mathbb{R}^{m+1}}{\text{minimize}} \quad & \frac{1}{2}\|x - y\|_2^2 + \mathbf{1}_\mathcal{F}(z) \\ \text{subject to} \quad & z = \begin{bmatrix} Ax + b \\ c^T x + d \end{bmatrix} =: Gx + h, \end{aligned} \tag{C.3}$$

where for brevity we define $G = \begin{bmatrix} A \\ c^T \end{bmatrix}$ and $h = \begin{bmatrix} b \\ d \end{bmatrix}$, and where $\mathbf{1}_\mathcal{F}$ denotes the indicator function for membership in the set $\mathcal{F}$.

We describe our fast solution technique for computing this projection, as well as our method for obtaining gradients through the solution.

### C.1 COMPUTING THE PROJECTION

We construct a fast solver for problem (C.3) using an accelerated projected dual gradient method. Specifically, define $\mu = \mathbb{R}^{m+1}$ as the dual variable on the equality constraint in Equation (C.3). The Lagrangian for this problem can then be written as

$$\mathscr{L}(x, z, \mu) = \frac{1}{2}\|x - y\|_2^2 + \mathbf{1}_\mathcal{F}(z) + \mu^T(z - Gx - h), \tag{C.4}$$

and the dual problem is given by $\max_\mu \min_{x,z} \mathscr{L}(x, z, \mu)$. To form the dual problem, we minimize the Lagrangian with respect to $x$ and $z$ as

$$\inf_{x,z} \mathscr{L}(x, z, \mu) = \inf_x \frac{1}{2}\left\{\|x - y\|_2^2 - \mu^T Gx\right\} + \inf_z \{\mu^T z + \mathbf{1}_\mathcal{F}(z)\} - \mu^T h. \tag{C.5}$$

We note that the first term on the right side is minimized at $x^\star(\mu) = y + G^T \mu$. Thus, we see that

$$\inf_x \frac{1}{2}\{\|x - y\|_2^2 - \mu^T Gx\} = -\frac{1}{2}\mu^T GG^T \mu - \mu^T Gy. \tag{C.6}$$

For the second term, denote $\mu = (\tilde{\mu}, s)$ and $z = (\tilde{z}, t)$. We can then rewrite this term as

$$\inf_z \{\mu^T z + \mathbf{1}_\mathcal{F}(z)\} = \inf_{t \ge 0}\, \inf_{\tilde{z}}\, \{t \cdot s + \tilde{\mu}^T \tilde{z} \mid \|\tilde{z}\|_2 \le t\}. \tag{C.7}$$

For a fixed $t \ge 0$, the above objective is minimized at $\tilde{z} = -t\tilde{\mu}/\|\tilde{\mu}\|_2$. (The problem is infeasible for $t < 0$.) Substituting this minimizer into (C.7) and minimizing the result over $t \ge 0$ yields

$$\inf_z \{\mu^T z + \mathbf{1}_\mathcal{F}(z)\} = \inf_{t \ge 0}\, t(s - \|\tilde{\mu}\|_2) = -\mathbf{1}_\mathcal{F}(\mu) \tag{C.8}$$

where the last identity follows from definition of the second-order cone $\mathcal{F}$. Hence the negative dual problem becomes

$$\underset{\mu}{\text{minimize}} \quad \frac{1}{2}\mu^T GG^T \mu + \mu^T (Gy + h) + \mathbf{1}_{\mathcal{F}}(\mu). \tag{C.9}$$

We now solve this problem via Nesterov's accelerated projected dual gradient method (Nesterov, 2013). For notational brevity, define $f(\mu) := \frac{1}{2}\mu^T GG^T \mu + \mu^T (Gy + h)$. Then, starting from arbitrary $\mu^{(-1)}, \mu^{(0)} \in \mathbb{R}^{m+1}$ we perform the iterative updates

$$\nu^{(k)} = \mu^{(k)} + \beta^{(k)}(\mu^{(k)} - \mu^{(k-1)})$$
$$\mu^{(k+1)} = \mathcal{P}_{\mathcal{F}}\left(\nu^{(k)} - \frac{1}{L_f}\nabla f(\nu^{(k)})\right), \tag{C.10}$$

where $L_f = \lambda_{\max}(GG^T)$ is the Lipschitz constant of $f$, and $\mathcal{P}_{\mathcal{F}}$ is the projection operator onto $\mathcal{F}$ (which has a closed form solution; see Bauschke (1996)). Letting $m_f = \lambda_{\min}(GG^T)$ denote the strong convexity constant of $f$, the momentum parameter is then scheduled as (Nesterov, 2013)

$$\beta^k = \begin{cases} \dfrac{k-1}{k+2} & \text{if } m_f = 0 \\[2mm] \dfrac{\sqrt{L_f} - \sqrt{m_f}}{\sqrt{L_f} + \sqrt{m_f}} & \text{if } m_f > 0. \end{cases} \tag{C.11}$$

After computing the optimal dual variable $\mu^\star$, i.e., the fixed point of (C.10), the optimal primal variable can be recovered via the equation $x^\star = y + G^T \mu^\star$ (as can be observed from the first-order conditions of the Lagrangian (C.4)).

## C.2 Obtaining Gradients

In order to incorporate the above projection into our neural network, we need to compute the gradients of all problem variables (i.e., $G$, $h$, and $y$) through the solution $x^\star$. In particular, we note that $x^\star$ has a direct dependence on both $G$ and $y$, and an indirect dependence on all of $G$, $h$, and $y$ through $\mu^\star$.

To compute the relevant gradients through $\mu^\star$, we apply the implicit function theorem to the fixed point of the update equations (C.10). Specifically, as these updates imply that $\mu^\star = \nu^\star$, their fixed point can be written as

$$\mu^\star = \mathcal{P}_{\mathcal{F}}\left(\mu^\star - \frac{1}{L_f}\nabla f(\mu^\star)\right). \tag{C.12}$$

Define $M := \frac{\partial \mathcal{P}_{\mathcal{F}}(\cdot)}{\partial(\cdot)}\big|_{(\cdot)=\mu^\star - \frac{1}{L_f}\nabla f(\mu^\star)}$, and note that $\nabla f(\mu^\star) = GG^T \mu^\star + Gy + h$. The differential of the above fixed-point equation is then given by

$$\mathrm{d}\mu^\star = M \times \left(\mathrm{d}\mu^\star - \frac{1}{L_f}\left(\mathrm{d}GG^T \mu^\star + G\mathrm{d}G^T \mu^\star + GG^T \mathrm{d}\mu^\star + \mathrm{d}Gy + G\mathrm{d}y + \mathrm{d}h\right)\right). \tag{C.13}$$

Rearranging terms to separate the differentials of problem outputs from problem variables, we see that

$$\left(I - M + \frac{1}{L_f}MGG^T\right)\mathrm{d}\mu^\star = -\frac{1}{L_f}M\left(\mathrm{d}GG^T \mu^\star + G\mathrm{d}G^T \mu^\star + \mathrm{d}Gy + G\mathrm{d}y + \mathrm{d}h\right), \tag{C.14}$$

where $I$ is the identity matrix of appropriate size.

As described in e.g. Amos and Kolter (2017), we can then use these equations to form the Jacobian of $\mu^\star$ with respect to any of the problem variables by setting the differential of the relevant problem variable to $I$ and of all other problem variables to 0; solving the resulting equation for $\mathrm{d}\mu^\star$ then yields the value of the desired Jacobian. However, as these Jacobians can be large depending on problem size, we rarely want to form them explicitly. Instead, given some backward pass vector $\frac{\partial \ell}{\partial \mu^\star} \in \mathbb{R}^{1 \times (m+1)}$ with respect to the optimal dual variable, we want to directly compute the gradient

of the loss with respect to the problem variables: e.g., for $y$, we want to directly form the result of the product $\frac{\partial \ell}{\partial \mu^\star} \frac{\partial \mu^\star}{\partial y} \in \mathbb{R}^{1 \times n}$. We do this via a similar method as presented in Amos and Kolter (2017), and refer the reader there for a more in-depth explanation of the method described below.

Define $J := I - M + \frac{1}{L_f} M G G^T$ to represent the coefficient of $\mathrm{d}\mu^\star$ on the left side of Equation (C.14). Given $\frac{\partial \ell}{\partial \mu^\star}$, we then compute the intermediate term

$$\mathrm{d}_\mu := -J^{-T} \left( \frac{\partial \ell}{\partial \mu^\star} \right)^T. \tag{C.15}$$

We can then form the relevant gradient terms directly as

$$\left( \frac{\partial \ell}{\partial \mu^\star} \frac{\partial \mu^\star}{\partial G} \right)^T = \frac{1}{L_f} M \left( \mathrm{d}_\mu (G^T \mu^\star)^T + \mu^\star (G^T \mathrm{d}_\mu)^T + \mathrm{d}_\mu y^T \right)$$

$$\left( \frac{\partial \ell}{\partial \mu^\star} \frac{\partial \mu^\star}{\partial h} \right)^T = \frac{1}{L_f} M \mathrm{d}_\mu \tag{C.16}$$

$$\left( \frac{\partial \ell}{\partial \mu^\star} \frac{\partial \mu^\star}{\partial y} \right)^T = \frac{1}{L_f} G^T M \mathrm{d}_\mu.$$

In these computations, we note that as our solver returns $x^\star$, the backward pass vector we are given is actually $\frac{\partial \ell}{\partial x^\star} \in \mathbb{R}^{1 \times n}$; thus, we compute $\frac{\partial \ell}{\partial \mu^\star} = \frac{\partial \ell}{\partial x^\star} \frac{\partial x^\star}{\partial \mu^\star} = \frac{\partial \ell}{\partial x^\star} G^T$ for use in Equation (C.15).

Accounting additionally for the direct dependence of some of the problem variables on $x^\star$ (recalling that $x^\star = y + G^T u^\star$), the desired gradients are then given by

$$\left( \frac{\partial \ell}{\partial G} \right)^T = \left( \frac{\partial \ell}{\partial x^\star} \frac{\partial x^\star}{\partial G} + \frac{\partial \ell}{\partial x^\star} \frac{\partial x^\star}{\partial u^\star} \frac{\partial u^\star}{\partial G} \right)^T = \mu^\star \frac{\partial \ell}{\partial x^\star} + \frac{1}{L_f} M \left( \mathrm{d}_\mu (G^T \mu^\star)^T + \mu^\star (G^T \mathrm{d}_\mu)^T + \mathrm{d}_\mu y^T \right)$$

$$\left( \frac{\partial \ell}{\partial h} \right)^T = \left( \frac{\cancelto{0}{\partial \ell}{\partial x^\star}}{\partial h} + \frac{\partial \ell}{\partial x^\star} \frac{\partial x^\star}{\partial u^\star} \frac{\partial u^\star}{\partial h} \right)^T = \frac{1}{L_f} M \mathrm{d}_\mu$$

$$\left( \frac{\partial \ell}{\partial y} \right)^T = \left( \frac{\partial \ell}{\partial x^\star} \frac{\partial x^\star}{\partial y} + \frac{\partial \ell}{\partial x^\star} \frac{\partial x^\star}{\partial u^\star} \frac{\partial u^\star}{\partial y} \right)^T = \left( \frac{\partial \ell}{\partial x^\star} \right)^T + \frac{1}{L_f} G^T M \mathrm{d}_\mu. \tag{C.17}$$

## D    WRITING THE CART-POLE PROBLEM AS AN NLDI

In the cart-pole task, our goal is to balance an inverted pendulum resting on top of a cart by exerting horizontal forces on the cart. Specifically, the state of this system is defined as $x = [p_x, \dot{p}_x, \varphi, \dot{\varphi}]^T$, where $p_x$ is the cart position and $\varphi$ is the angular displacement of the pendulum from its vertical position; we seek to stabilize the system at $x = \vec{0}$ by exerting horizontal forces $u \in \mathbb{R}$ on the cart. For a pendulum of length $\ell$ and mass $m_p$, and for a cart of mass $m_c$, the dynamics of the system are (as described in Tedrake (2009)):

$$\dot{x} = \begin{bmatrix} \dot{p}_x \\ \frac{u + m_p \sin \varphi (\ell \dot{\varphi}^2 - g \cos \varphi)}{m_c + m_p \sin^2 \varphi} \\ \dot{\varphi} \\ \frac{(m_c + m_p) g \sin \varphi - u \cos \varphi - m_p \ell \dot{\varphi}^2 \cos \varphi \sin \varphi}{l(m_c + m_p \sin^2 \varphi)} \end{bmatrix}, \tag{D.1}$$

where $g = 9.81 \text{ m/s}^2$ is the acceleration due to gravity. We rewrite this system as an NLDI by defining $\dot{x} = f(x, u)$ and then linearizing the system about its equilibrium point as

$$\dot{x} = \mathrm{J}_f(0,0) \begin{bmatrix} x \\ u \end{bmatrix} + I_n w, \quad \|w\| \le \|Cx + Du\|, \tag{D.2}$$

where $J_f$ is the Jacobian of the dynamics, $w = f(x, u) - J_f(0, 0) [x \quad u]^T$ is the linearization error, and $I_n$ is the $n \times n$ identity matrix. We bound this linearization error by numerically obtaining the matrices $C$ and $D$, assuming that $x$ and $u$ are within a neighborhood of the origin. We describe this process in more detail below. As a note, while we employ an NLDI here to characterize the linearization error, it is also possible to characterize this error via polytopic uncertainty (see Appendix J); we choose to use an NLDI here as it yields a much smaller problem description than a PLDI in this case.

### D.1 DERIVING $J_f(0, 0)$

For $\dot{x} = f(x, u)$, we see that

$$
J_f(x, u) = \begin{bmatrix} 0 & 1 & 0 & 0 & 0 \\ 0 & 0 & \partial\ddot{p}_x/\partial\varphi & \partial\ddot{p}_x/\partial\dot{\varphi} & \partial\ddot{p}_x/\partial u \\ 0 & 0 & 0 & 1 & 0 \\ 0 & 0 & \partial\ddot{\varphi}/\partial\varphi & \partial\ddot{\varphi}/\partial\dot{\varphi} & \partial\ddot{\varphi}/\partial u, \end{bmatrix},
\tag{D.3}
$$

where

$$
\frac{\partial\ddot{p}_x}{\partial\varphi} = \frac{m_p \cos\varphi \left(\dot{\varphi}^2 l - g\cos\varphi\right) + g m_p \sin^2\varphi}{m_c + m_p \sin^2\varphi} - \frac{2 m_p \sin\varphi\cos\varphi \left(m_p \sin\varphi \left(\dot{\varphi}^2 l - g\cos\varphi\right) + u\right)}{\left(m_c + m_p \sin^2\varphi\right)^2},
$$

$$
\frac{\partial\ddot{p}_x}{\partial\dot{\varphi}} = \frac{2\dot{\varphi} l m_p \sin\varphi}{m_c + m_p \sin^2\varphi},
$$

$$
\frac{\partial\ddot{p}_x}{\partial u} = \frac{1}{m_c + m_p \sin^2\varphi},
$$

$$
\frac{\partial\ddot{\varphi}}{\partial\varphi} = \frac{\begin{matrix} g(m_c + m_p)\cos\varphi + \dot{\varphi}^2 l m_p \sin^2\varphi \\ -\dot{\varphi}^2 l m_p \cos^2\varphi + u\sin\varphi \end{matrix}}{l\left(m_c + m_p\sin^2\varphi\right)} - \frac{\begin{matrix} 2m_p \sin\varphi\cos\varphi(g(m_c + m_p))\sin\varphi \\ -\dot{\varphi}^2 l m_p \sin\varphi\cos\varphi - u\cos\varphi \end{matrix}}{l\left(m_c + m_p\sin^2\varphi\right)^2},
$$

$$
\frac{\partial\ddot{\varphi}}{\partial\dot{\varphi}} = \frac{-2\dot{\varphi} m_p \sin\varphi\cos\varphi}{m_c + m_p \sin^2\varphi},
$$

$$
\frac{\partial\ddot{\varphi}}{\partial u} = \frac{-\cos\varphi}{l(m_c + m_p\sin^2\varphi)}.
$$

We thus see that

$$
J_f(0, 0) = \begin{bmatrix} 0 & 1 & 0 & 0 & 0 \\ 0 & 0 & -m_p g/m_c & 0 & 1/m_c \\ 0 & 0 & 0 & 1 & 0 \\ 0 & 0 & g(m_c + m_p)/l m_c & 0 & -1/m_c \end{bmatrix}.
\tag{D.4}
$$

### D.2 OBTAINING $C$ AND $D$

We then seek to construct matrices $C$ and $D$ that bound the linearization error $w$ between the true dynamics $\dot{x}$ and our first-order linear approximation $J_f(0, 0) \begin{bmatrix} x \\ u \end{bmatrix}$. To do so, we bound the error of this approximation entry-wise: that is, for each entry $i = 1, \ldots, s$, we want to find $F_i$ such that for all $x$ in some region $\underline{x} \le x \le \bar{x}$, and all $u$ in some region $\underline{u} \le u \le \bar{u}$,

$$
w_i^2 = \left(\nabla f_i(0) \begin{bmatrix} x \\ u \end{bmatrix} - \dot{x}_i\right)^2 \le \begin{bmatrix} x \\ u \end{bmatrix}^T F_i \begin{bmatrix} x \\ u \end{bmatrix}.
\tag{D.5}
$$

Then, given the matrix

$$
M = \begin{bmatrix} F_1^{T/2} & F_2^{T/2} & F_3^{T/2} & F_4^{T/2} & F_5^{T/2} & F_6^{T/2} \end{bmatrix}^T
\tag{D.6}
$$

we can then obtain $C = M_{1:s}$ and $D = M_{s:s+m}$ (where the subscripts indicate column-wise indexing).

We solve separately for each $F_i$ to minimize the difference between the right and left sides of Equation (D.5) (while enforcing that the right side is larger than the left side) over a discrete grid of points within $\underline{x} \le x \le \bar{x}$ and $\underline{u} \le u \le \bar{u}$. By assuming that $F_i$ is symmetric, we are able to cast this as a linear program in the upper triangular entries of $F_i$.

To obtain the matrices $C$ and $D$ used for the cart-pole experiments in the main paper, we let $\bar{x} = [1.5 \quad 2 \quad 0.2 \quad 1.5]^T$, $\bar{u} = 10$, $\underline{x} = -\bar{x}$, and $\underline{u} = -\bar{u}$. As each entry-wise difference in Equation (D.5) contained exactly three variables (i.e., a total of three entries from $x$ and $u$), we solved each entry-wise linear program over a mesh grid of 50 points per variable.

## E    WRITING QUADROTOR AS AN NLDI

In the planar quadrotor setting, our goal is to stabilize a quadcopter in the two-dimensional plane by controlling the amount of force provided by the quadcopter's right and left thrusters. Specifically, the state of this system is defined as $x = [p_x \quad p_z \quad \varphi \quad \dot{p}_x \quad \dot{p}_z \quad \dot{\varphi}]^T$, where $(p_x, p_z)$ is the position of the quadcopter in the vertical plane and $\varphi$ is its roll (i.e., angle from the horizontal position); we seek to stabilize the system at $x = \vec{0}$ by controlling the amount of force $u = [u_r, u_l]^T$ from right and left thrusters. We assume that our action $u$ is additional to a baseline force of $[mg/2 \quad mg/2]^T$ provided by the thrusters by default to prevent the quadcopter from falling. For a quadrotor with mass $m$, moment-arm $\ell$ for the thrusters, and moment of inertia $J$ about the roll axis, the dynamics of this system are then given by (as modified from Singh et al. (2020)):

$$\dot{x} = \begin{bmatrix} \dot{p}_x \cos\varphi - \dot{p}_z \sin\varphi \\ \dot{p}_x \sin\varphi + \dot{p}_z \cos\varphi \\ \dot{\varphi} \\ \dot{p}_z \dot{\varphi} - g\sin\varphi \\ -\dot{p}_x \dot{\varphi} - g\cos\varphi + g \\ 0 \end{bmatrix} + \begin{bmatrix} 0 & 0 \\ 0 & 0 \\ 0 & 0 \\ 0 & 0 \\ 1/m & 1/m \\ \ell/J & -\ell/J \end{bmatrix} u, \tag{E.1}$$

where $g = 9.81$ m/s$^2$. We linearize this system via a similar method as for the cart-pole setting, i.e., as in Equation (D.2). We describe this process in more detail below. We note that since the dependence of the dynamics on $u$ is linear, we have that $D = 0$ for our resultant NLDI. As for cart-pole, while we employ an NLDI here to characterize the linearization error, it is also possible to characterize this error via polytopic uncertainty (see Appendix J); we choose to use an NLDI here as it yields a much smaller problem description than a PLDI in this case.

### E.1    DERIVING $\mathrm{J}_f(0,0)$

For $\dot{x} = f(x, u)$, we see that

$$\mathrm{J}_f(x, u) = \begin{bmatrix} 0 & 0 & -\dot{p}_x\sin\varphi - \dot{p}_z\cos\varphi & \cos\varphi & -\sin\varphi & 0 & 0 \\ 0 & 0 & \dot{p}_x\cos\varphi - \dot{p}_z\sin\varphi & \sin\varphi & \cos\varphi & 0 & 0 \\ 0 & 0 & 0 & 0 & 0 & 1 & 0 \\ 0 & 0 & -g\cos\varphi & 0 & \dot{\varphi} & \dot{p}_z & 0 \\ 0 & 0 & g\sin\varphi & -\dot{\varphi} & 0 & -\dot{p}_x & 0 \\ 0 & 0 & 0 & 0 & 0 & 0 & 0 \end{bmatrix}, \tag{E.2}$$

and thus

$$\mathrm{J}_f(0,0) = \begin{bmatrix} 0 & 0 & 0 & 1 & 0 & 0 & 0 \\ 0 & 0 & 0 & 0 & 1 & 0 & 0 \\ 0 & 0 & 0 & 0 & 0 & 1 & 0 \\ 0 & 0 & -g & 0 & 0 & 0 & 0 \\ 0 & 0 & 0 & 0 & 0 & 0 & 0 \\ 0 & 0 & 0 & 0 & 0 & 0 & 0 \end{bmatrix}. \tag{E.3}$$

### E.2    OBTAINING $C$ AND $D$

We obtain the matrices $C$ and $D$ via a similar method as described in Appendix D, though in practice we only consider the linearization error with respect to $x$ (i.e., since the dynamics are linear with respect to $u$, we have $D = 0$). We let $\bar{x} = [1 \quad 1 \quad 0.15 \quad 0.6 \quad 0.6 \quad 1.3]$ and $\underline{x} = -\bar{x}$. As for cart-pole, each entry wise difference in the equivalent of Equation (D.5) contained exactly three variables (i.e., a total of three entries from $x$ and $u$), and each entry-wise linear program was solved over a mesh grid of 50 points per variable.

## F    DETAILS ON THE MICROGRID SETTING

For our experiments, we build upon the microgrid setting given in Lam et al. (2016). In this system, the state $x \in \mathbb{R}^3$ captures voltage deviations, frequency deviations, and the amount of power generated by a diesel generator connected to the grid; the action $u \in \mathbb{R}^2$ describes the current associated with a storage device and a solar PV inverter; and the disturbance $w \in \mathbb{R}$ describes the difference between the amount of power demanded and the amount of power produced by solar panels on the grid. The authors also define a performance index $y \in \mathbb{R}^2$ which captures voltage and frequency deviations (i.e., two of the entries of the state $x$).

To construct an NLDI of the form (3) for this system, we directly use the $A$, $B$, and $G$ matrices given in Lam et al. (2016). We generate $C$ i.i.d. from a normal distribution and let $D = 0$, to represent the fact that the disturbance $w$ and the entries of the state $x$ are correlated, but that $w$ is likely not correlated with the actions $u$. Finally, we let $Q$ and $R$ be diagonal matrices with 1 in the entries corresponding to quantities represented in the performance index $y$, and with 0.1 in the rest of the diagonal entries, to emphasize that the variables in $y$ are the most important in describing the performance of the system.

## G    GENERATING AN ADVERSARIAL DISTURBANCE

In the NLDI settings explored in our experiments, we seek to construct an "adversarial" disturbance $w(t)$ that obeys the relevant norm bounds $\|w(t)\|_2 \leq \|Cx(t) + Du(t)\|_2$ while maximizing the loss. To do this, we use a model predictive control method where the actions taken are $w(t)$. Specifically, for each policy $\pi$, we model $w(t)$ as a neural network specific to that policy. Every 10 steps of a roll-out, we optimize $w(t)$ through gradient descent to maximize the loss over a horizon of 40 steps, subject to the constraint $\|w(t)\|_2 \leq \|Cx(t) + Du(t)\|_2$.

## H    ADDITIONAL EXPERIMENTAL DETAILS

**Initial states.** To pick initial states in our experiments, for the synthetic settings, we sample each attribute of the state i.i.d. from a standard Gaussian distribution. For cart-pole and planar quadrotor, we sample uniformly from bounds chosen such that the non-robust LQR algorithm (under the original dynamics) did not go unstable. For cart-pole, these bounds were chosen to be $p_x \in [-1, 1]$, $\varphi \in [-0.1, 0.1]$, $\dot{p}_x = \dot{\varphi} = 0$. For planar quadrotor, these bounds were $p_x, p_z \in [-1, 1]$, $\varphi \in [-0.05, 0.05]$, $\dot{p}_x = \dot{p}_z = \dot{\varphi} = 0$.

**Constructing NLDI bounds.** Given these initial states, for the cart-pole and quadrotor settings, we needed to construct our NLDI disturbance bounds such that they would hold over the *entire* trajectory of the robust policy; if not, the robustness specification (A.8) would not hold, and our agent might in fact increase the Lyapunov function. To ensure this approximately, we used a simple heuristic: we ran the (non-robust) LQR agent for a full episode with 50 different starting conditions, and constructed an $L_\infty$ ball around all states reached in any of these trajectories. We then used these $L_\infty$ balls on the states to construct the matrices $C$ and $D$ for our disturbance bounds, using the procedure described in Appendices D and E.

**Computing infrastructure and runtime.** All experiments were run on an XPS 13 laptop with an Intel i7 processor. The planar quadrotor and synthetic NLDI experiment with $D = 0$ took about 1 day to run (since the projections were simple half-space projections), while all the other synthetic domains and cart-pole took about 3 days to run. The majority of the run-time was in computing the adversarial disturbances for test-time evaluations.

**Hyperparameter selection.** For our experiments, we did not perform large parameter searches. The learning rate we chose for our model-based planner, (both robust and non-robust) remained constant for the different domains; we tried learning rates of $1 \times 10^{-3}, 1 \times 10^{-4}, 1 \times 10^{-5}$ and found $1 \times 10^{-3}$ worked best for the non-robust version and $1 \times 10^{-4}$ worked best for the robust version. For our PPO hyperparameters, we simply used those used in the original PPO paper.

One parameter we had to tune for each environment was the time step. In particular, we had to pick a time step high enough that we could run episodes for a reasonable total length of time (within which the non-robust agents would go unstable), but low enough to reasonably approximate a continuous-time setting (since, for our robustness guarantees, we assume the agent's actions evolve in continuous time). Our search space was small, however, consisting of 0.05, 0.02, 0.01, and 0.005 seconds.

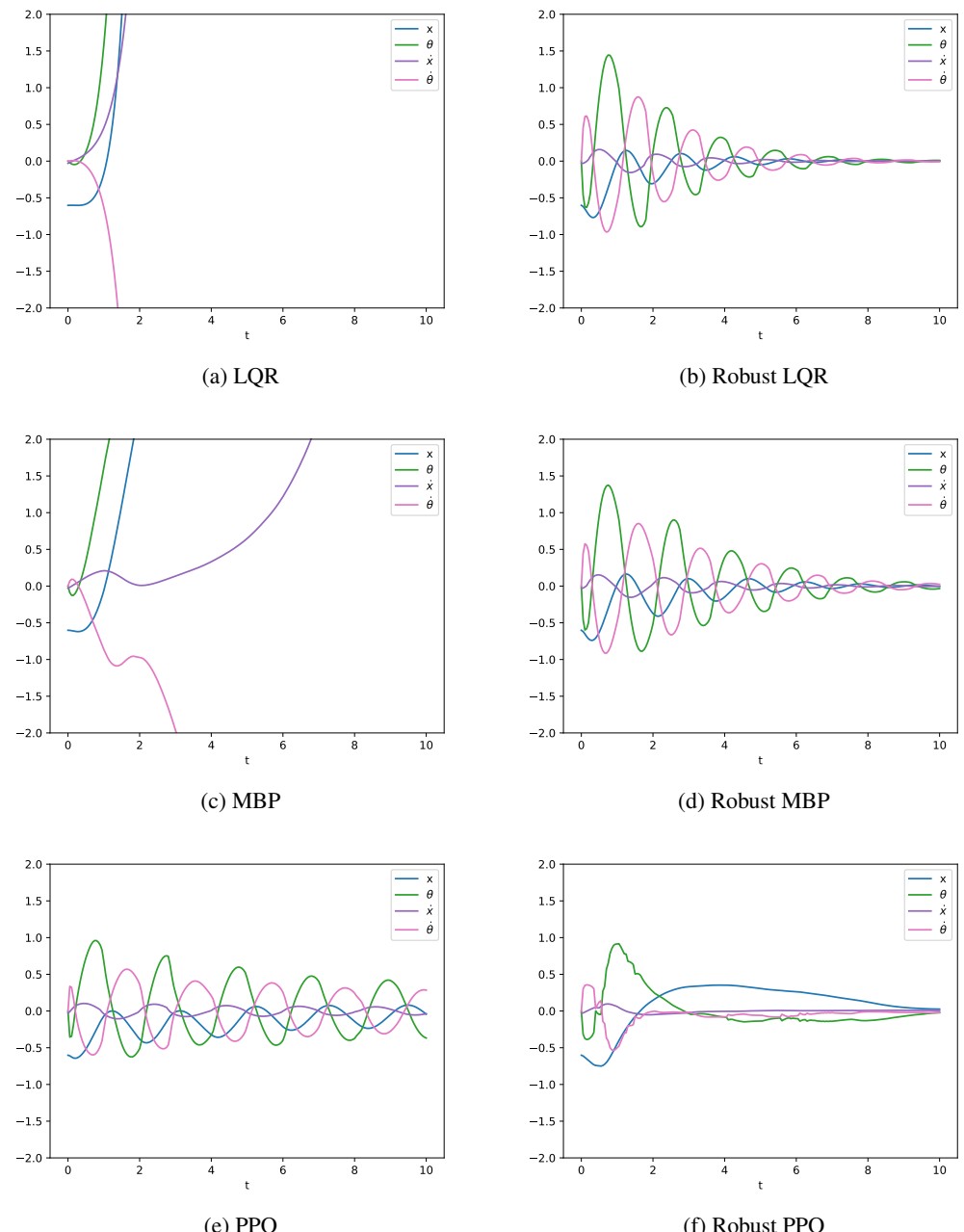

Figure H.1: Trajectories of 6 different methods on the cart-pole domain under adversarial dynamics.

**Trajectory plots.** Figure H.1 shows sample trajectories of different methods in the cart-pole domain under adversarial dynamics. The non-robust LQR and model-based planning approaches both diverge and the non-robust PPO doesn't diverge, but doesn't clearly converge after 10 seconds. The robust methods, on the other hand, all clearly converge after 10 seconds.

**Runtime comparison.** Tables H.1 and H.2 show the evaluation and training time of our methods and the baselines over 50 episodes run in parallel. In the NLDI cases where $D = 0$, i.e., Generic NLDI ($D = 0$) and Quadrotor, our projection adds only a very small computational cost. In the other cases, the additional computational cost is more significant, but our method is still far less expensive than the Robust MPC method.

| Environment | LQR | MBP | PPO | Robust LQR | Robust MPC | RARL | Robust MBP* | Robust PPO* |
|---|---|---|---|---|---|---|---|---|
| Generic NLDI ($D = 0$) | 0.63 | 0.61 | 0.84 | 0.57 | 718.06 | 0.71 | 0.73 | 0.94 |
| Generic NLDI ($D \neq 0$) | 0.64 | 0.62 | 0.83 | 0.58 | 824.86 | 0.81 | 15.13 | 25.38 |
| Cart-pole | 0.55 | 0.67 | 0.84 | 0.53 | 646.90 | 0.84 | 10.12 | 13.37 |
| Quadrotor | 0.95 | 0.98 | 1.19 | 0.88 | 3348.68 | 1.14 | 1.15 | 1.30 |
| Microgrid | 0.58 | 0.61 | 0.79 | 0.57 | 601.90 | 0.74 | 8.14 | 10.25 |
| Generic PLDI | 0.57 | 0.54 | 0.76 | 0.51 | 819.24 | 0.73 | 69.35 | 64.03 |
| Generic $H_\infty$ | 0.84 | 0.80 | 1.03 | 0.76 | N/A | 1.00 | 47.81 | 63.67 |

Table H.1: Time (in seconds) taken to run each method on the test set of every environment for 50 episodes run in parallel.

| Environment | MBP | PPO | RARL | Robust MBP* | Robust PPO* |
|---|---|---|---|---|---|
| Generic NLDI ($D = 0$) | 26.36 | 101.77 | 102.37 | 30.78 | 114.60 |
| Generic NLDI ($D \neq 0$) | 26.46 | 100.79 | 82.53 | 221.35 | 1158.28 |
| Cart-pole | 25.49 | 87.04 | 98.90 | 146.34 | 689.93 |
| Quadrotor | 41.24 | 131.48 | 112.95 | 46.13 | 159.06 |
| Microgrid | 23.03 | 112.52 | 87.71 | 113.61 | 436.64 |

Table H.2: Time (in minutes) taken to train each method in every environment.

# I  EXPERIMENTS FOR PLDIS AND $H_\infty$ CONTROL SETTINGS

In addition to the NLDI settings explored in the main text, we test the performance of our method on PLDI and $H_\infty$ control settings. As for the experiments in the main text, we choose a time discretization based on the speed at which the system evolves, and run each episode for 200 steps over this discretization. In both cases, we use a randomly generated LQR objective where the matrices $Q^{1/2}$ and $R^{1/2}$ are drawn i.i.d. from a standard normal distribution.

**Synthetic PLDI setting.** We generate PLDI instances (A.9) with $s = 5$, $a = 3$, and $L = 3$. Specifically, we generate convex hull matrices $(A_1, B_1), \ldots, (A_3, B_3)$ i.i.d. from normal distributions, and generate $(A(t), B(t))$ by using a randomly-initialized neural network with softmax output to weight the convex hull matrices. Episodes were run for 2 seconds at a discretization of 0.01 seconds.

**Synthetic $H_\infty$ setting.** We generate $H_\infty$ control instances (A.11) with $s = 5$, $a = 3$, and $d = 2$ by generating matrices $A$, $B$ and $G$ i.i.d. from normal distributions. The disturbance $w(t)$ was produced using a randomly-initialized neural network, with its output scaled to satisfy the $\mathcal{L}_2$ bound on the disturbance. Specifically, we scaled the output of the neural network to satisfy an attenuating norm-bound on the disturbance; at time $t$, the norm-bound was given by $20 \times f(2 \times t/T)$, where $T$ is the time horizon and $f$ is the standard normal PDF function. Episodes were run for $T = 2$ seconds at a discretization of 0.01 seconds.

Results are given in Figure I.1 and Table I.1.

| Environment | | LQR | MBP | PPO | Robust LQR | Robust MPC | RARL | Robust MBP* | Robust PPO* |
|---|---|---|---|---|---|---|---|---|---|
| Generic PLDI | O | 96.3 | **3.3** | 8.0 | 19.2 | 19.2 | 15.8 | 18.6 | **10.2** |
| | A | ——— | *unstable* | ——— | 43.3 | 44.1 | *unstable* | 21.9 | 16.1 |
| Generic $H_\infty$ | O | 181 | **88** | 114 | 165 | N/A | **115** | 116 | 125 |
| | A | 219 | 112 | 143 | 206 | N/A | 145 | 147 | 158 |

Table I.1: Performance of various approaches, both robust (right) and non-robust (left), on domains of interest. We report average quadratic loss over 50 episodes under the original dynamics (O) and under an adversarial disturbance (A). For the original dynamics (O), the best performance for both non-robust methods and robust methods is in bold (lower loss is better). We use "*unstable*" to indicate cases where the relevant method became unstable. Our robust methods (denoted by *) improve performance over Robust LQR in the average case, while remaining stable under adversarial dynamics, whereas the non-robust methods either went unstable or received much larger losses.

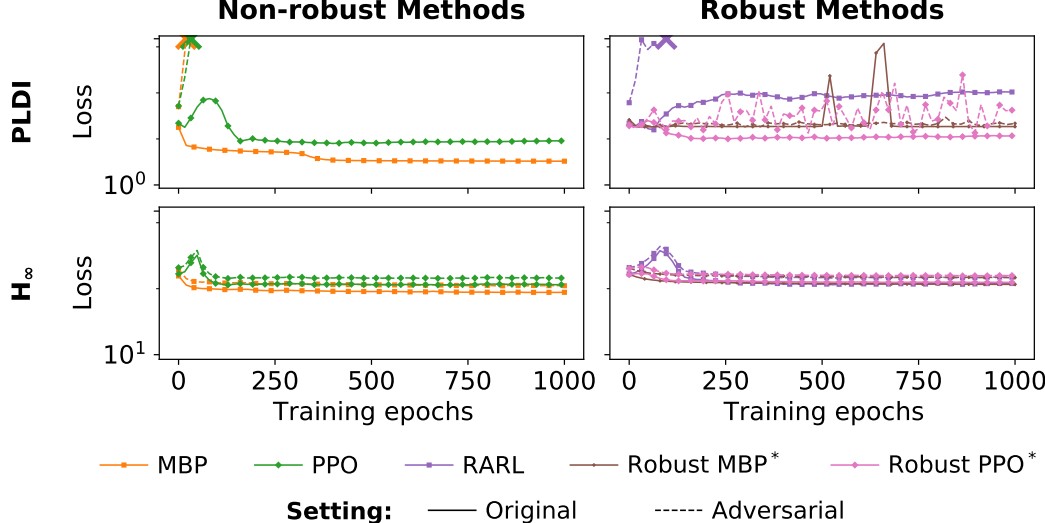

Figure I.1: Representative results for our experimental settings. For each training epoch (10 updates for the MBP model and 18 for PPO), we report average quadratic loss over 50 episodes, and use "X" to indicate cases where the relevant method became unstable. (Lower loss is better.) Our robust methods (denoted by *) improve performance over Robust LQR in the average case, while (unlike the non-robust methods) remaining stable under adversarial dynamics throughout the training process.

## J  NOTES ON LINEARIZATION VIA PLDIS AND NLDIS

While we linearize the cart-pole and quadrotor dynamics via NLDIs in our experiments, we note that these dynamics can also be characterized via PLDIs. More generally, in this section, we show how we can use the framework of PLDIs to model linearization errors arising in the analysis of nonlinear systems.

Consider the nonlinear dynamical system

$$\dot{x} = f(x, u) \text{ with } f(0, 0) = 0. \tag{J.1}$$

for $x \in \mathbb{R}^s$ and $u \in \mathbb{R}^a$. Define $\xi = (x, u)$. We would like to represent the above system as a PLDI in the region $\mathcal{R} := \{\xi \mid \underline{\xi} \leq \xi \leq \bar{\xi}\}$ including the origin. The mean value theorem states that for each component of $f$, we can write

$$f_i(\xi) = f_i(0) + \nabla f_i(z)^T \xi, \tag{J.2}$$

for some $z = t\xi$, where $t \in [0, 1]$. Now, let $p = s + a$. Defining the Jacobian of $f$ as

$$\mathrm{J}_f(z) = \begin{bmatrix} \nabla f_1(z)^T \\ \vdots \\ \nabla f_p(z)^T \end{bmatrix}, \tag{J.3}$$

and recalling that $f(0) = 0$, we can rewrite (J.2) as

$$f(\xi) = \mathrm{J}_f(z)\xi. \tag{J.4}$$

Now, suppose we can find component-wise bounds on the matrix $\mathrm{J}_f(z)$ over $\mathcal{R}$, i.e,

$$\underline{M} \leq \mathrm{J}_f(z) \leq \bar{M} \text{ for all } z \in \mathcal{R}. \tag{J.5}$$

We can then write

$$\mathrm{J}_f(z) = \sum_{1 \leq i,j \leq p} m_{ij}(t) E_{ij} \quad \text{with} \quad m_{ij}(t) \in [\underline{m}_{ij}, \bar{m}_{ij}], \tag{J.6}$$

where $E_{ij} = e_i e_j^T$ and $e_i$ is the $i$-th unit vector in $\mathbb{R}^p$.

We now seek to bound the Jacobian using polytopic bounds. To do this, note that we can write

$$\mathrm{J}_f(z) = \sum_{\kappa=1}^{2^{p^2}} \gamma_\kappa A_\kappa \quad \gamma_\kappa \geq 0, \ \sum_\kappa \gamma_\kappa = 1, \tag{J.7}$$

where $A_\kappa$'s are the vertices of the polytope in (J.6), i.e.,

$$A_\kappa \in \mathcal{V} = \left\{ \sum_{1 \leq i,j \leq p} m_{ij} E_{ij} \mid m_{ij} \in \{\underline{m}_{ij}, \bar{m}_{ij}\} \right\}. \tag{J.8}$$

Together, Equations (J.2), (J.4), (J.7), and (J.8) characterize the original nonlinear dynamics as a PLDI.

We note that this PLDI description is potentially very large; in particular, the size of $\mathcal{V}$ is exponential in the square of the number of non-constant entries in the Jacobian $\mathrm{J}_f(z)$, which could be as large as $2^{p^2} = 2^{(s+a)^2}$. This problem size may therefore become intractable for larger control settings.

We note, however, that we can in fact express this PLDI more concisely as an NLDI. More precisely, we would like to find matrices $A, B, C$ parameterizing the form of NLDI below, which is equivalent to that presented in Equation (3) (see Chapter 4 of Boyd et al. (1994)):

$$Df(z) \in \{A + B\Delta C \mid \|\Delta\|_2 \leq 1\} \quad \text{for all } z \in \mathcal{R}. \tag{J.9}$$

It can shown that the solution to the SDP

$$
\begin{aligned}
&\text{minimize } \ \mathrm{tr}(V + W) \\
&\text{subject to } \ W \succ 0 \\
&\qquad\qquad \begin{bmatrix} V & (A_\kappa - A)^T \\ A_\kappa - A & W \end{bmatrix} \succeq 0, \ \forall A_\kappa \in \mathcal{V}
\end{aligned}
\tag{J.10}
$$

yields the matrices $A$, $B$, and $C$ with $V = C^T C$ and $W = BB^T$, which can be used to construct NLDI (J.9). While the NLDI here is more concise than the PLDI, the trade-off is that the NLDI norm bounds obtained via this method may be rather loose. As such, for our settings, we obtain NLDI bounds numerically (see Appendices D and E), as these are tighter than NLDI specifications obtained via the above method (though they are potentially slightly inexact). An alternative approach would be to examine how to tighten the conversion from PLDIs to NLDIs, which has been explored in other work (e.g. Kuiava et al. (2013)).

