# OpenReview forum: "Enforcing robust control guarantees within neural network policies"
_ICLR.cc/2021/Conference — ICLR 2021 Poster_

### Official Review · AnonReviewer4 · 2020-10-28
**This paper proposed a class of nonlinear control policies that combines the expressiveness of neural networks with the provable stability guarantees of traditional robust control, under the area of safe RL. It was claimed that such new policies has the main advantages to good performance in average case than traditional robust control technique, and has the advantage of provably robustness guarantee over the  traditional non-robust RL methods.**

**Rating:** 6
**Confidence:** 4

**Review:**

Pros:
  [1] The problem of RL policies with both robustness guarantee and good average performance is interesting and useful for many practical applications.
  [2] Paper is well-written and clear.
  [3] The proposed method/approach is inspiring and novel, an the results look promising for this proposed new method.


Main concern:
  The results presented in the experiments section is not very comprehensive and not convincing enough to justify the claimed benefits of the proposed approach v.s. traditional robust LQR approach. In particular, as we could see from Table 1, although in the original dynamics scenario, the proposed robust MBP and robust PPO approach has better performance than the robust LQR, in the adversarial disturbance scenarios,  the proposed robust MBP and robust PPO might perform significantly worse (e.g., about 8 times larger loss for Microgrid case) than the traditional robust LQR, which implies that, for many scenarios that are not worst case, the proposed method could perform worse than the robust LQR (i.e., from 8 times larger loss (e.g., 7.12 v.s. 0.86) in the adversarial case, to  slightly smaller loss (0.61 v.s. 0.73) in the original case, there is a large gap there). And there is not a clear and well justified criteria in the paper to clarify, for most real world applications, what disturbance is defined as normal(/average/original) case, and what scenario is for adversarial case, and also how about the disturbance between these two scenarios? Due to above concerns, and also note that the examples provided in this paper is very limited rather than comprehensive, it is not convincing enough to claim the overall performance benefits of the proposed approach over the traditional robust control techniques. More comprehensive experimental studies and evidence are needed to well justify the claimed performance benefits.

Edit: upgrade the rating to 6 with the clarifications from the authors, with which the submission is clearer and more convincing now.

---

> ### Author Response · Authors · 2020-11-18
> **Response to R4**
>
> Thank you for your thoughtful questions and comments. We would like to clarify our experimental setup in the hopes that this will address some of your concerns. We have also made substantial revisions to Section 5 of the paper in order to make these particular points clearer.
>
> In our experiments, we aim to test two things:
> * How well does each method perform in the _standard_ or _average-case_ setting in which it is used? (We evaluate this by directly comparing LQR costs under the original dynamics.)
> * Is the method able to withstand large disturbances, i.e., still keep the system stable in worst-case scenarios? (We evaluate this by examining whether or not the LQR loss “blows up” for a particular method under adversarial dynamics.)
>
> Given this, our experimental design was to construct one case that is more representative of expected settings (original dynamics), and to construct one illustrative case designed explicitly to show that the non-robust methods “blow up” in certain scenarios where the robust methods remain stable (adversarial dynamics).
>
> In particular, our adversarial dynamics were created by computing a worst-case disturbance for every model at every time step, using model-predictive control. As such, the losses reported under adversarial dynamics represent the absolute “worst-case” losses that any method might experience if it were explicitly targeted.The adversarial setting does not necessarily represent a real-world dynamical setting (in particular, it requires that the disturbance be tailored specifically to be as destructive as possible to the model being used), but was merely constructed to illustrate a point.
>
> Given this setup, we believe the “original dynamics” are certainly more representative of the average-case settings in which each method might usually be used, and as such, the performance in this original setting is more indicative of “usual-case” performance than performance on the adversarial case is.
>
> (Some minor additions to our approach, such as training simultaneously on multiple sets of dynamics, e.g., in a multi-task learning setting, could also be used to _ensure_ good performance on particular sets of dynamics. This could be an interesting direction for future work.)
>
> Regarding the number of experiments: We note that we constructed all experimental domains used in this paper from scratch, and thus prioritized settings that were illustrative of the kinds of systems that might be considered under the dynamical models we study. In particular, unlike for RL, there are no standard sets of benchmarks or test suites for robust control problems (as far as we are aware), and the process of converting from RL-style setups to robust control-style setups can be somewhat onerous. However, if there are standard sets of robust control benchmarks (for e.g., LMI or NLDI settings) on which we can conduct additional experiments, we would be eager to be given pointers to these benchmarks.

---

> > ### Author Response · Authors · 2020-11-19
> > **Further clarification**
> >
> > We discussed your comment further, and wanted to elaborate on our previous response. To clarify, you’re absolutely correct that there are situations where Robust LQR will outperform our method under some disturbance models that satisfy the given bounds. However, the setting we’re interested in, and maybe we should have emphasized this further, is the case where the agent is given access to a chunk of training data from the system (either online or offline).
> >
> > If that data is drawn from random runs of the system, then our method will empirically outperform Robust LQR in the average case. (This is the case we are trying to simulate in our experiments on the “original dynamics.”)
> >
> > However, if there is a random run that differs from these average dynamics (within the given bounds), our system is guaranteed to not go unstable. (This is the case we are trying to illustrate via our tests on “adversarial dynamics.”)
> >
> > Please let us know if you have additional questions, or if it would be helpful for us to clarify further.

---

### Official Review · AnonReviewer1 · 2020-10-30
**Interesting stability-guaranteed neural control method through introducing convex optimization layers**

**Rating:** 6
**Confidence:** 4

**Review:**

In this paper, a neural control method is proposed with stability guarantees. The control is assumed to be from a neural network that takes in the state. Stability is guaranteed by projecting the control to the set that satisfies the Lyapunov stability condition for the LQR problem. In particular, minimizing the cost of LQR cost subject to stability constraints can be cast as an SDP for norm-bouned linear differential inclusions. Through making use of the convex optimization layers proposed in Agrawal et al. (2019), the SDP can be added as a layer after the neural policy and efficient projections can be derived such that implicit function theorem can be utilized to differentiate through the fixed point (the optimal conditions of the SDP), such that end to end learning is possible. The proposed approach is compared with the unconstrained method on various tasks.  Both model-based and model-free RL algorithms are used as the neural policy for comparison. The stability-guaranteed approach is able to remain stable even under bounded adversarial dynamics. In comparison, the non-robust methods fail to maintain stability.

In general, I like the idea of enforcing stability by introducing the convex optimization layer. Although the dynamics model used is still relatively basic, but the nice convex formulation provides an opportunity to incorporate stability certificates to neural policy that enable end-to-end training. The stability is roughly maintained as illustrated in the experiments. However, it would be better if trajectories can be visualized in some way to show that the proposed method can stabilize the system. One potential concern of the method is that it would be more computationally expensive than the unconstrained method, therefore it is of interest to compare the running time.

Some minor points regarding the experiments: the description of the methods used are not properly defined or referenced (such as PPO, RARL, MBP). Also, the experiments for PLDIs and H-infinity control settings seem to be missing (only description of the setup is found).

----------------------After author's response--------------------

The response addressed most of my concerns and included experiments results of trajectory visualization and run-time comparison. I think the paper would be an interesting contribution to the conference.

---

> ### Author Response · Authors · 2020-11-18
> **Response to R1**
>
> Thank you for your comments and suggestions! To respond to your suggestions and clarify some additional points:
>
> > it would be better if trajectories can be visualized in some way to show that the proposed method can stabilize the system
>
> Thank you for this suggestion. We have now included visualizations of the trajectories produced by different methods for the cart-pole domain in Appendix H of our paper.
>
> > One potential concern of the method is that it would be more computationally expensive than the unconstrained method, therefore it is of interest to compare the running time.
>
> We are currently running timing results, and plan to include them in the next revision of the paper.
>
> > the description of the methods used are not properly defined or referenced (such as PPO, RARL, MBP)
>
> Good point. We have now included more thorough descriptions of/references to these methods (as well as an additional baseline method we now include, called Robust MPC) in section 5.2 of the paper.
>
> > the experiments for PLDIs and H-infinity control settings seem to be missing (only description of the setup is found)
>
> The results are in Figure I.1 and Table I.1 in Appendix I, but we had actually forgotten to include references to these figures within the text of Appendix I. We have now added those text references.
>
> > Through making use of the convex optimization layers proposed in Agrawal et al. (2019)
>
> To clarify, while cvxpylayers is a general-purpose option that can be used within our framework, it can be somewhat computationally expensive to employ given its generality. As a result, it can be important to develop faster, special-purpose solvers for use in particular settings. As such, one of the contributions of this work is actually in developing a custom efficient, differentiable SDP projection layer that we employ in the NLDI and $H_\infty$ settings. This layer uses an accelerated projected dual gradient method for the forward pass, and we derive gradients for the backward pass via implicit differentiation through the fixed point equations of this solution technique. The high-level ideas used in deriving this layer are similar to those in Agrawal et al. (2019), but the details are somewhat different.

---

> > ### Author Response · Authors · 2020-11-20
> > **Follow up on experiment times**
> >
> > To follow up, we have now included timing results for our methods in Appendix H of the paper. We see that in the NLDI cases where D=0, i.e., Generic NLDI (D=0) and Quadrotor, our projection adds only a very small computational cost. In the other cases, the additional computational cost is more significant, but our method is still far less expensive than the Robust MPC method. Additional speedup could be achieved by reducing the precision of the solvers that we use to compute the projections.

---

### Official Review · AnonReviewer2 · 2020-11-03
**The rebuttal responses most of my concerns, I think it is a good combination of RL and robust control**

**Rating:** 6
**Confidence:** 4

**Review:**

In this paper, the authors proposed a new robust controller design approach, in which the controller is parameterised by DNN. They show that by integrating custom convex-optimization-based projection layers into a nonlinear policy, they can construct a provably robust neural network policy class.


1.This is paper is heavily in control theory. The proof of Theorem 1 and corollary 1 is standard in the sense of robust control and LMI. The key contribution I can see is Section 4.3 on deriving differential projections.

2.In control community, adaptive dynamic programming is exactly to deal with the problem proposed in the paper. Frank Lewis and many others had done a lot of work in this field. I saw the authors cited one of his papers in 2006 in the section of RL. Unfortunately, in comparison with the classic and recent progress in this field is missing.

3.The definition of stability and exponential stability is not given, which learning community may not be aware of.

4.A key reference is missing “H∞ Model-free Reinforcement Learning with Robust Stability Guarantee, arXiv:1911.02875”. In this paper, the authors propose a model-free approach to learning the Lyapunov function and policy simultaneously. I think this paper is more general than this ICLR submission. A remark and comparison are needed.

5.Regarding the experiment, I didn’t see how the nonlinear systems are converted to (1). If I assume this is possible, how such approximation gap can be quantified and how will this gap affect the theoretical results.

6.One important detail in the experiment is missing: how is the initial condition selected. Let’s take the cart pole as an example: if the initial condition is (very) close to the equilibrium point, the nonlinearities will be minimal. I don’t think robust LQR will be worse. While RL is outstanding in the nonlinear region, the authors should make a fair comparison/add more scenarios.

7.Comparison with PPO and is MBP is not proper and unfair. PPO and MBP is a data-based method, i.e., a concrete system model is NOT needed. While the proposed method must need a model and the theoretical result depends on the model parameters

---

> ### Author Response · Authors · 2020-11-18
> **Response to R2 - part 1**
>
> Thank you for your thorough review of our work. We believe, however, that there might be some misunderstanding about the main setting (NLDIs) that our work addresses, and its key differences from the $H_\infty$ settings that the cited works in your review address. In particular, these differences mean that much of the theory from this previous $H_\infty$ work cannot be directly applied to the NLDI setting.
>
> Specifically, in $H_\infty$ settings, disturbances are assumed to have finite energy (but otherwise be unstructured), and as a result, the optimal control policy is linear (as derived from the game algebraic Riccati equation (GARE) for linear systems). However, in the NLDI setting we address, this is not the case. For NLDIs, disturbances are structured and need not have finite energy, and can in fact depend arbitrarily on the state $x$ and action $u$. As such, __the optimal control policy is not necessarily linear__. As a result, the kinds of approaches that are derived in $H_\infty$ settings cannot be imported into NLDI settings, in particular because many $H_\infty$ works depend on the linearity of the optimal controller in their derivations.
>
> In addition, the forms of guarantees considered in these settings are also different: $H_\infty$ settings seek to bound the $\mathcal{L}_2$ gain of the disturbance-to-output map, and guarantee asymptotic stability only for the disturbance-free case. In contrast, in NLDI settings, we seek to guarantee asymptotic stability in general (i.e., even in the presence of the disturbance). The details of the approaches that can be used differ greatly on this basis as well.
>
> As such, we emphasize that while the prior work you mention is conceptually relevant (and we have updated the paper to include some additional discussion), the assumptions and (consequently) the results in that work are in many ways orthogonal to the setting addressed here.
>
> With that said, to respond to your individual points:

---

> > ### Author Response · Authors · 2020-11-18
> > **Response to R2 - part 2**
> >
> > > 1.This is paper is heavily in control theory. The proof of Theorem 1 and corollary 1 is standard in the sense of robust control and LMI. The key contribution I can see is Section 4.3 on deriving differential projections.
> >
> > The novelty of our contribution is in exploiting existing tools from robust control within the context of reinforcement learning, which is enabled by the use of differentiable optimization. While seemingly simple, this is a powerful paradigm that both (a) brings robustness guarantees from control theory into reinforcement learning, and (b) brings the power of nonlinear control policies (learned via deep reinforcement learning) into robust control settings. As such, we do not believe our contribution should be penalized for using existing proof techniques, as implementation of these techniques in a manner that is usable by RL is non-trivial.
> >
> > We agree that deriving efficient, differentiable projections is an important part of this framework, and have now added additional sentences in the introduction and methods sections to further emphasize this part of the contribution.
> >
> > > 2.In control community, adaptive dynamic programming is exactly to deal with the problem proposed in the paper. Frank Lewis and many others had done a lot of work in this field. I saw the authors cited one of his papers in 2006 in the section of RL. Unfortunately, in comparison with the classic and recent progress in this field is missing.
> >
> > As discussed above, the body of work on adaptive dynamic programming has mainly focused on the case that the disturbance is finite-energy but otherwise unstructured. In contrast, in our setting, we are assuming that the disturbance term (coming from modeling errors, for example) satisfies some known structural bounds that can be exploited to design stable deep nonlinear policies that can do better than linear controllers while maintaining stability. As such, the theory associated with much of the literature on adaptive dynamic programming is not directly applicable to our setting.
> >
> > Our work is closer to the literature on robust model-predictive control in terms of assumptions. However, robust MPC approaches are often limited to linear controllers. We have now added a comparison to robust MPC in the revised paper, and show that our method outperforms this approach in terms of average-case performance.
> >
> > > 3.The definition of stability and exponential stability is not given, which learning community may not be aware of.
> >
> > Good point. In the revised manuscript, we have added some additional context around Equation (2) to provide intuition for the meaning of stability, and to clarify that Equation (2) characterizes exponential stability. We have also added a more formal definition of exponential stability in a footnote.
> >
> > > 4. A key reference is missing “H∞ Model-free Reinforcement Learning with Robust Stability
> > Guarantee, arXiv:1911.02875”. In this paper, the authors propose a model-free approach to learning the Lyapunov function and policy simultaneously. I think this paper is more general than this ICLR submission. A remark and comparison are needed.
> >
> > This paper is relevant, but we disagree with the reviewer that this paper is more general than our submission, in particular because it addresses a different setting. Specifically, the mentioned paper addresses the $H_\infty$ setting, whose differences from our current setting we have already described. In addition, the mentioned paper addresses mean-squared stability in a stochastic setting, whereas ours addresses deterministic stability under worst-case realizations of uncertainty. As such, the method proposed in that paper cannot be directly applied to the setting we consider. Although we currently fix the Lyapunov function (given by a robust linear controller) and learn the policy, we agree that simultaneously learning both of these would be an interesting future direction for NLDIs.
> >
> > While we had referenced this paper in our previous version, we have now added some additional discussion of this paper in the related work.

---

> > > ### Author Response · Authors · 2020-11-18
> > > **Response to R2 - part 3**
> > >
> > > > 5.Regarding the experiment, I didn’t see how the nonlinear systems are converted to (1). If I assume this is possible, how such approximation gap can be quantified and how will this gap affect the theoretical results.
> > >
> > > > 6.One important detail in the experiment is missing: how is the initial condition selected. Let’s take the cart pole as an example: if the initial condition is (very) close to the equilibrium point, the nonlinearities will be minimal. I don’t think robust LQR will be worse. While RL is outstanding in the nonlinear region, the authors should make a fair comparison/add more scenarios.
> > >
> > > As referenced in Section 5.1, we provide details on how the cart-pole and quadrotor tasks are converted to the NLDI (3) in Appendices D and E, respectively. To summarize, we linearize these systems by deriving the Jacobians of the dynamics, and then numerically derive norm bounds on the linearization error that hold within a particular region around equilibrium.
> > >
> > > Importantly, we only consider initialization points in regions where these bounds are guaranteed to hold, i.e., where the nonlinearities are adequately described by our system model. We had a sentence about this initialization scheme in the main text of the previous version (with additional details in the appendix), but have now expanded upon this at the end of Section 5.2 to make it clearer.
> > >
> > > To summarize, for cart-pole and quadrotor, we use a heuristic to identify an $L_\infty$ ball, $B_{\text{init}}$, around the equilibrium, from which we can draw initialization points. $B_{\text{init}}$ is constructed such that it is fully contained within a level set of the robust LQR Lyapunov function, which in turn is fully contained within the region where our linearization holds. As such, any method that is guaranteed to decrease this Lyapunov function (namely, Robust LQR and our robust methods) will never produce a trajectory that leaves the linearization region when given an initialization point within $B_{\text{init}}$.
> > >
> > > (As the synthetic NLDI and synthetic microgrid settings we consider have true dynamics that follow the NLDI model (3), we do not need to convert these systems, and so can simply draw random initial states from $x \in \mathbb{R}^s.$)
> > >
> > > In sum, this means that within our experiments, we test _precisely_ those settings where Robust LQR should excel — i.e., draw initialization points in a region around the equilibrium where the system model holds throughout all robust trajectories — and find that our neural network-based methods _still_ outperform Robust LQR in these settings.

---

> > > > ### Author Response · Authors · 2020-11-18
> > > > **Response to R2 - part 4/4**
> > > >
> > > > > 7.Comparison with PPO and is MBP is not proper and unfair. PPO and MBP is a data-based method, i.e., a concrete system model is NOT needed. While the proposed method must need a model and the theoretical result depends on the model parameters
> > > >
> > > > We believe there is a fundamental misunderstanding here. The claim is that comparison to MBP and PPO is not fair because these are “pure” RL strategies that use no model of the system at all. However, this is the entire point, and precisely the key limitation of RL (in the NLDI setting) that our paper tries to address.
> > > >
> > > > In particular, our method proposes to address a setting where we do _not_ know the true underlying model, but we _do_ know a robust control specification for the model. (Critically, this is not the same as knowing the full model, because a number of different unmodeled dynamics can be captured by the disturbance term.)
> > > >
> > > > The fact that RL and planning methods like MBP and PPO cannot accommodate robust control specifications is precisely the key limitation that our paper tries to address: These methods are not able to make provable guarantees on their performance _because_ they do not take these bounds on the dynamics into account, _even when these bounds are available_.  The whole point of our approach is that if we are given a setting we _do_ have such a control specification, then we can combine deep RL with this control specification to both:
> > > > * Achieve provably robust performance, in contrast to “pure” MBP and PPO methods that do not incorporate the control specification and therefore cannot guarantee stability. (Indeed, “pure” MBP/PPO perform better under the original dynamics — and pure MBP in fact has full knowledge of these dynamics, given that it is a planning rather than an RL method — but the relevance of this comparison is to show that they can go unstable without a control specification.)
> > > > * Perform _better_ than traditional robust control methods that construct a linear control law, because the data-driven nature of MBP and PPO will adapt the nonlinear control policy to the specific settings of the true underlying dynamics. (In the previous version, the primary comparison here was to Robust LQR. We have now also added a comparison to Robust MPC.)
> > > >
> > > > Therefore, we disagree that comparisons to pure RL/planning methods are unfair, but rather that RL and planning methods of this sort have previously been limited in terms of robustness guarantees (and that this limitation has not to date been addressed in NLDI settings). Our proposed method aims to address exactly this problem, by incorporating control specifications into methods like MBP and PPO. We fully agree this gives our algorithm more information over “pure” MBP or PPO, but indeed, that is the point here, as this additional information allows our method to perform _better_ than traditional robust linear control, while simultaneously remaining stable under worst case perturbations.

---

### Official Review · AnonReviewer3 · 2020-11-05
**Interesting formulation of robust control to learn robust control with NN**

**Rating:** 6
**Confidence:** 3

**Review:**

The paper combines robust control theory with NNs to obtain robustness guarantees. The paper shows stable behavior on adversarial dynamic models on many different simulated tasks. The paper's main idea revolves around projecting the output of a NN policy to be within the set of a Lyapunov function (stability condition) (e.g., exponential stability). I have some concerns which are listed below,

1. What are the limitations of the approach for larger dimensional control problems, such as a 7-dof arm? It's unclear if the method can scale to large dimensional control problems.

2. Fig.1 methods are hard to separate, plotting with qualitatively different colors would be helpful. Additionally, explaining the horizontal lines that start before 0 would be useful.

3. While the results show stable behavior, the loss in the adversarial setting  doesn't improve during training. Is there some reasoning for this? I am suspecting the adversarial dynamics is randomizing after every epoch? Showing the method improving with a fixed adversarial dynamics over epochs would be good to show improvement in addition to stability.

---

> ### Author Response · Authors · 2020-11-18
> **Response to R3**
>
> Thank you for your questions and comments! Please see our responses below.
>
> > 1. What are the limitations of the approach for larger dimensional control problems, such as a 7-dof arm? It's unclear if the method can scale to large dimensional control problems.
>
> We expect that our method will scale gracefully to larger control problems. In particular, the main computational expense of our method is in the projections onto the stable sets of actions, and the cost of these projections scales polynomially with the state-action size. (For comparison, GPs, e.g., scale exponentially.)
>
> The reason that we did not test on more or larger domains is simply that these domains are difficult to construct from scratch. In particular, unlike for RL, there are no standard sets of benchmarks or test suites for robust control problems (as far as we are aware), and the process of converting from RL-style setups to robust control-style setups can be somewhat onerous. As such, we focused on converting a limited set of existing RL domains to our control theory setup (see Appendices D and E for our conversions of cart-pole and quadrotor, respectively) in order to demonstrate the efficacy of our method.
>
> > 2. Fig.1 methods are hard to separate, plotting with qualitatively different colors would be helpful. Additionally, explaining the horizontal lines that start before 0 would be useful.
>
> We have now removed some methods from the plot in Figure 1 in order to make the plot easier to read. In particular, the plot shows the evolving performance of the different methods on the test set as they train; however, since the LQR methods are not learning methods, their performance on the test set does not change over time. As such, we have now removed these non-learning methods from Figure 1 in order to de-clutter the plot, but still include their test-time performance in Table 1.
>
> (This is why the horizontal lines for those LQR methods previously started before 0 — we were simply plotting a horizontal line with these methods’ test-time performance, and were not careful about where this line started. Our apologies for that.)
>
> > 3. While the results show stable behavior, the loss in the adversarial setting doesn't improve during training. Is there some reasoning for this? I am suspecting the adversarial dynamics is randomizing after every epoch? Showing the method improving with a fixed adversarial dynamics over epochs would be good to show improvement in addition to stability.
>
> We would like to further clarify our experimental setup (and have made substantial revisions to Section 5.2 of the paper in order to do this). In particular, all methods are trained on the original dynamical systems (described in Section 5.1), and then tested on both (a) these original dynamics, and (b) adversarial dynamics.
>
> These adversarial dynamics are computed _separately for each model at each point in time_. In particular, we construct an adversarial disturbance using model-predictive control to try to maximize the loss associated with a given model at any given time, and we evolve each model’s associated adversarial disturbance over time as the model evolves.
>
> With that background, to answer your questions, the loss in the adversarial setting does not improve over time both (a) because the models are not explicitly optimizing for this adversarial setting (they are optimizing for the original setting), and (b) because the adversarial dynamics are computed to be as destructive as possible to each model at each point in time (i.e., the adversarial plot shows the “worst case” loss over time). Similarly, fixing any one set of adversarial dynamics would not make sense, given that these dynamics must evolve over time (and be unique to each model) in order to be truly adversarial. (The setting of “fixing” particular dynamics and seeing the models evolve over time to optimize these dynamics is what is addressed by our experiments on the original dynamics.)

---

> > ### Comment · AnonReviewer3 · 2020-11-20
> > **Revised experimental details are helful**
> >
> > Thanks for discussing the experimental setup. The figure is also much easier to read now.

---

### Decision · Program_Chairs · 2021-01-07
**Final Decision**

**Decision:**

Accept (Poster)

**Comment:**

While the reviewers seem to like the main idea of the work, they had several concerns, particularly regarding the experiments (both their setup and description) and the overall language of the paper that they found it more suitable for the control community than the ML and representation learning community. The authors provided very long response and tried to address the issues raised by the reviewers during the rebuttals. Fortunately, the response addressed some of the issues they raised and now they all see the paper marginally above the line. However, reading the reviews and response shows that the paper can highly benefit from better writing and describing the experiments. So, I would strongly recommend that the authors include all the information they provided for the reviewers during the rebuttal phase in the paper and improve its quality.